# Prevalent chromosome fusion in *Vibrio cholerae* O1

**Aline Cuénod** [1] ✉, **Denise Chac** [2], **Ashraful I. Khan**[3], **Fahima Chowdhury**[3], **Randy W. Hyppa**[4], **Susan M. Markiewicz**[2], **Amelia Rice**[2], **Akhil Kholwadwala**[5], **Stephen B. Calderwood** [6,7], **Edward T. Ryan**[6,7,8], **Jason B. Harris**[7,9,10], **Regina C. LaRocque**[6,7], **Taufiqur R. Bhuiyan**[3], **Gerald R. Smith** [4], **Firdausi Qadri** [3], **Patrick Lypaczewski** [1] ✉, **Ana A. Weil** [2,11] ✉ & **B. Jesse Shapiro** [1,12,13] ✉

Two circular chromosomes are a defining feature of the bacterial family *Vibrionaceae*, including the pathogen *Vibrio cholerae*, with rare reports of isolates with a single, fused chromosome. Here, we use long-read sequencing to analyse 467 *V. cholerae* O1 isolates from 47 cholera patients and household contacts in Bangladesh. We identify several independent chromosome fusion events that are likely transmissible within a household. Fusions occur in a 12 kilobase-pair homologous sequence shared between the two chromosomes and are stable for at least 200 generations under laboratory conditions. We find no detectable effect of fusion on *V. cholerae* growth, virulence factor expression, or biofilm formation. The factors promoting fusion, affecting chromosome stability, and subtle phenotypic or clinical consequences merit further investigation.

Cholera is a waterborne infectious disease affecting millions of people yearly and causing outbreaks where sanitary infrastructure is inadequate[1]. The ongoing seventh cholera pandemic is caused by a pathogenic lineage (7PET) of *Vibrio cholerae* O1 carrying four virulence-associated genomic islands: VPI-1, VPI-2, VSP-I and VSP-II[2–4]. In Bangladesh, where cholera is endemic, the 7PET sublineage BD2 was dominant between 2009 and 2018, followed by BD1.2, which was responsible for a large outbreak in Dhaka in 2022[5]. *V. cholerae* typically carries two chromosomes: the larger ~3 megabase-pair (Mbp) chromosome 1 and the smaller ~1 Mbp chromosome 2. When chromosome 2 replication is impaired under laboratory conditions, the two chromosomes can fuse to restore cell division[6]. Out of thousands of sequenced genomes, only three *V. cholerae* with fused chromosomes have been reported to date from natural environments[6–9]. These have typically been considered rare exceptions to the bipartite genome structure. However, due in part to limitations of short-read sequencing, the prevalence of chromosome fusion in *V. cholerae* remains unknown.

Here, we observe chromosome fusion in 58 out of 467 clinical *V. cholerae* isolates collected from 47 patients living in 21 households in Dhaka, Bangladesh. We identify multiple independent fusion events that appear to be stable enough to be transmitted within households. Fusions occur in a 12-kilobase-pair homologous sequence shared between the two chromosomes and are stable for 200 generations under laboratory conditions. We find no detectable effect of fusion on *V. cholerae* growth, virulence factor expression, or biofilm formation.

[1]Department of Microbiology and Immunology, McGill University, Montréal, QC, Canada. [2]Department of Medicine, University of Washington, Seattle, WA, USA. [3]International Centre for Diarrhoeal Disease Research, Bangladesh, (ICDDR, B), Dhaka, Bangladesh. [4]Division of Basic Sciences, Fred Hutchinson Cancer Center, Seattle, WA, USA. [5]Department of Biology, McGill University, Montréal, QC, Canada. [6]Division of Infectious Diseases, Massachusetts General Hospital, Boston, MA, USA. [7]Department of Medicine, Harvard Medical School, Boston, MA, USA. [8]Department of Immunology and Infectious Diseases, Harvard T.H. Chan School of Public Health, Boston, MA, USA. [9]Department of Pediatrics, Harvard Medical School, Boston, MA, USA. [10]Division of Global Health, Massachusetts General Hospital for Children, Boston, MA, USA. [11]Department of Global Health, University of Washington, Seattle, WA, USA. [12]McGill Genome Centre, McGill University, Montréal, QC, Canada. [13]McGill Centre for Microbiome Research, McGill University, Montréal, QC, Canada. ✉e-mail: aline@cuenod.email; patrick.lypaczewski@mcgill.ca; anaweil@uw.edu; jesse.shapiro@mcgill.ca

More subtle phenotypic effects or clinical impacts of chromosome fusions remain to be investigated.

## Results and discussion
### Chromosome fusion is prevalent in clinical *V. cholerae* O1 isolates
We aimed to detect chromosome fusion in clinical *V. cholerae* isolates and identify potential fusion mechanisms. To do so, we used long-read nanopore sequencing of 467 *V. cholerae* isolates, collected between 2015 and 2018 from 47 patients (21 index cases and 26 household contacts) from 21 households in Dhaka, Bangladesh (Fig. 1A, Supplementary Data 1, Fig. S1). All isolates were identified as serotype O1. Of these, 409 genomes assembled into two circular chromosomes (3 and 1 Mbp each) and 58 into a single 4 Mbp chromosome. All 58 single-chromosome genomes resulted from an apparent fusion of chromosomes 1 and 2. These fused chromosomes were identified in ten different people from five different households. These people include four index cases and six household contacts, two of whom experienced cholera symptoms (Supplementary Data 1).

### No phenotypic effect of chromosome fusion identified
Comparing closely related fused and non-fused isolates from the same patient (*n* = 2 pairs, 0 and 2 high-quality SNVs between genomes within a pair; Methods) we found no difference in growth or in expression of

the key virulence factors cholera toxin (*ctx*) and toxin co-regulated pilus (*tcp*), or in their capability to form biofilms (Fig. S2). This does not exclude the possibility of subtle phenotypic differences that could be detected with larger sample sizes or with other assays.

### Chromosome fusion is confirmed using Pulsed-Field Gel Electrophoresis
As independent verification of chromosome fusion, we subjected a subset of isolates to pulsed-field gel electrophoresis (PFGE). We included one putative fused-chromosome isolate per patient for which at least one fused chromosome was assembled (*n* = 10) and three putative non-fused-chromosome isolates for comparison. As expected, we detected one band at 4 Mbp for all putative fused-chromosome isolates and two bands at 3 and 1 Mbp for the non-fused isolates, corresponding to the known sizes of chromosomes 1 and 2, respectively (Fig. 1B). Some of the fused isolates have a weak, poorly defined band at about 1 Mbp in addition to the band at 4 Mbp, but the lack of a band at 3 Mbp in these isolates indicates that fusion did occur. We hypothesize the weak band to arise from fragmented DNA (<1 Mbps) which ran to this position in the pulsed field gel. Chromosome fusion thus does not appear to be an artefact of sequencing or assembly.

### Chromosome fusions occur at 12 Kbp homologous sequence
To understand the mechanism of fusion, we scanned the flanking regions on either side of the fusion site. We found that in fused

**A**

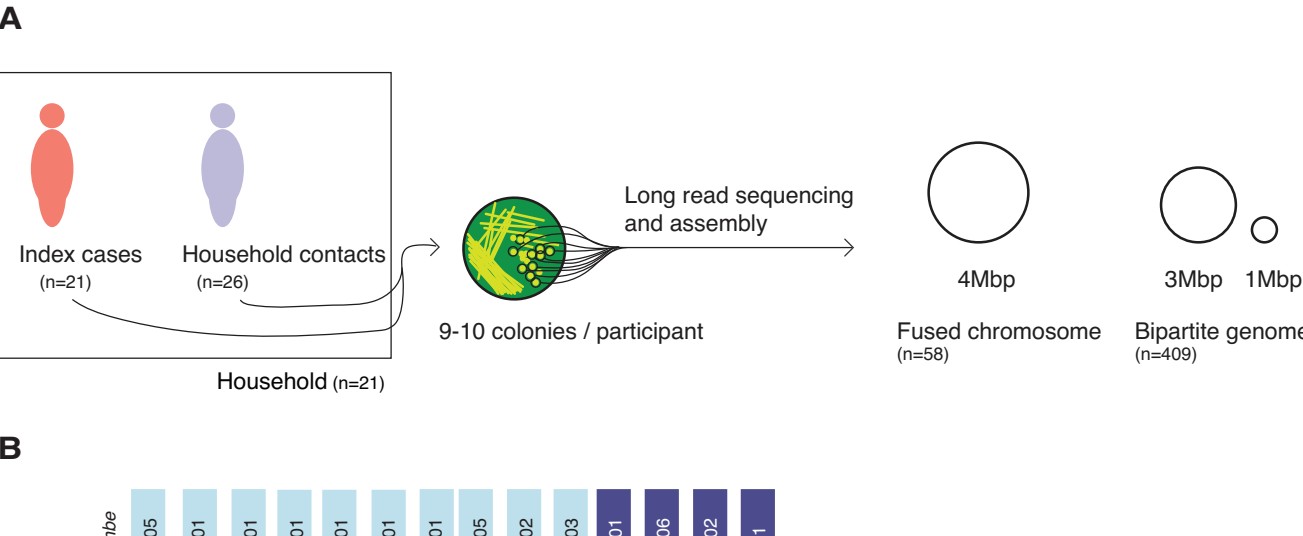

**B**

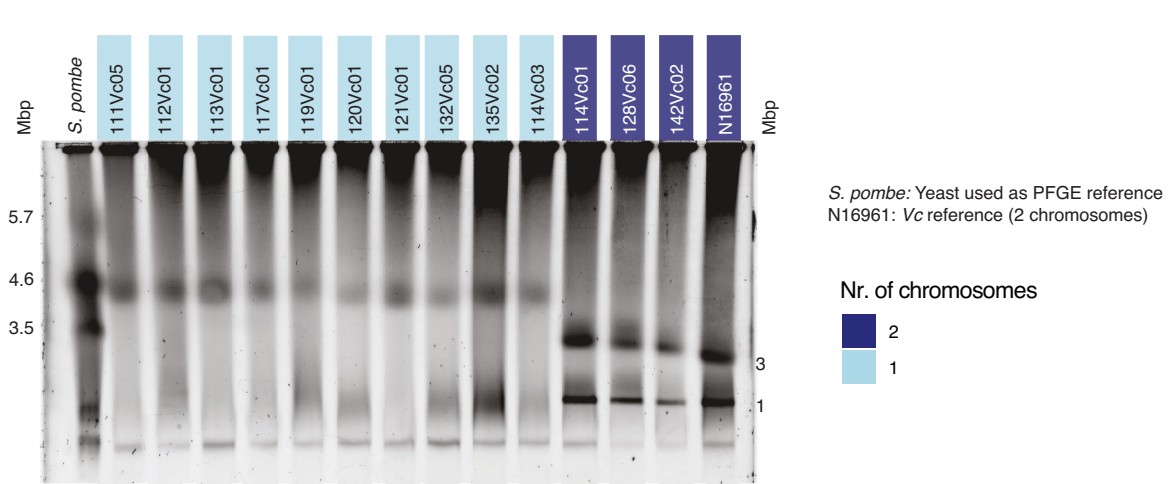

**Fig. 1 | *V. cholerae* with a fused chromosome identified in multiple patients and households. A** Schematic representation of the study design; **B** PFGE results for the putatively fused (light blue) and non-fused (dark blue) chromosomes identified by sequencing. *Schizosaccharomyces pombe* was used as a DNA size marker and *V.*

*cholerae* strain N16961 as a well-characterized isolate with two chromosomes. PFGE was conducted twice. PFGE: Pulsed-Field Gel Electrophoresis; Mbp: Mega base pair; *Vc*: *Vibrio cholerae*; *S. pombe*: *Schizosaccharomyces pombe*. Source data is provided as a Source Data file.

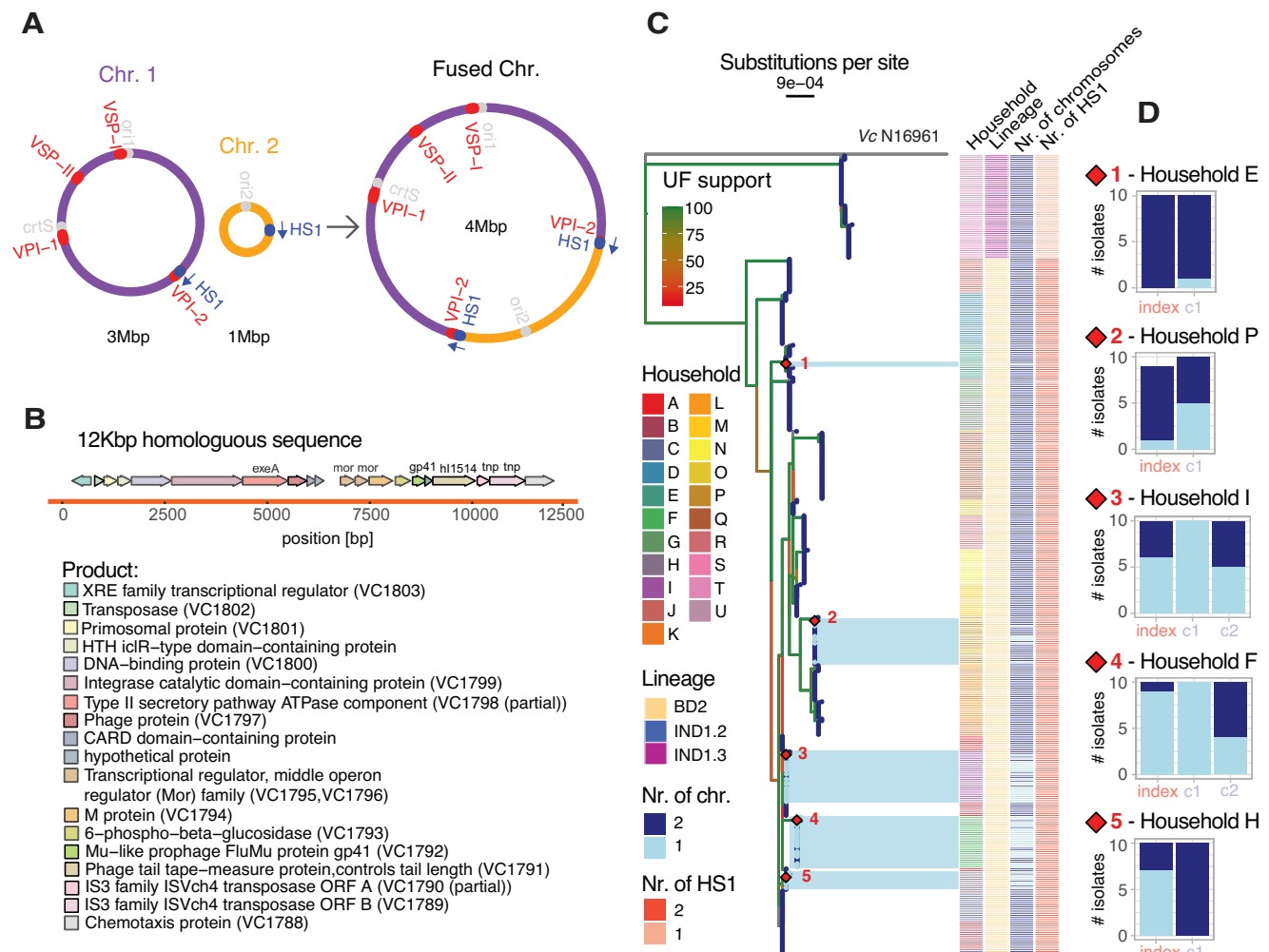

**Fig. 2 | Chromosome fusion events occur in *V. cholerae* sublineages with a shared homologous sequence on each chromosome. A** Schematic representation of the chromosome fusion with the location of HS1, *crtS*, the origins of replications and pathogenicity islands indicated. Chr. 2 does in rare cases carry an additional copy of VSP-I; (**B**) Annotation of genes within the homologous sequence (HS1) at the fusion site; (**C**) Phylogenetic tree based on 281 high-quality SNVs, along with household membership, the 7PET sublineage designation, the number of chromosomes identified by sequencing, and the number of times HS1 was detected in the genome. Highlighted in light blue are five branches or clades (numbered in red) that include genomes with a fused chromosome. The red diamonds in the tree depict their parent nodes. Branches are color-coded according to ultrafast bootstrap (UF) approximation support; (**D**) Distribution of fused and non-fused chromosome genomes amongst index cases and household contacts for all five households in which at least one fusion was detected. Mbp: Mega base pair; Kbp: Kilo base pair; Chr: Chromosome; HS1: Homologous Sequence 1; *Vc*: *Vibrio cholerae*; UF: UltraFast bootstrap approximation. Source data are provided as a Source Data file.

chromosomes, the chromosome 2 sequence is flanked by a 12 kilobase-pair (Kbp) homologous sequence (HS1) oriented in the same direction on either side of the integrated chromosome 1 sequence. In many non-fused strains, HS1 appears twice: once on chromosome 1 and once on chromosome 2 (Fig. 2A). This suggests homologous recombination at HS1 as a potential fusion mechanism. To further support the observations of fusion in the assemblies, we screened the raw sequence data for reads spanning HS1 and its flanking regions, which were highly concordant with the results of the assemblies (Fig. S3). On chromosome 1, HS1 is located within VPI-2 and contains genes VC1788-VC1803 whereas on chromosome 2 HS1 occurs between VCA0451 and VCA0453. HS1 encodes multiple proteins linked to horizontal gene transfer (Fig. 2B). These include two ISVch4 elements of the IS3 family, which have previously been identified as fusion sites in *V. cholerae* strains which fused upon experimental deletion of the DNA adenine methylase Dam[6] and in a naturally fused *V. cholerae* O1 isolate from a patient who travelled to Indonesia in 1997[8]. These are not the same recombination sequences or IS3 elements that have previously been described in two

other naturally fused non-O1 strains[7,9], indicating multiple viable fusion mechanisms.

## Multiple independent fusion events appear to be stable enough to be transmitted

We next examined the phylogenetic distribution of fused and non-fused chromosomes. The phylogenetic clustering of patients within a household is consistent with previous studies[10,11] suggesting *V. cholerae* transmission within households (Fig. 2C), including instances of potential fused chromosome transmission in households P, I, and F (Fig. 2D). All fused chromosomes were part of the 7PET sublineage BD2 (Fig. 2C). Most BD2 genomes in our dataset contained two copies of HS1 (one on each chromosome), likely explaining their propensity for fusion. By contrast, other sublineages – notably IND1.3 – contained only one HS1 copy (Fig. 2C), preventing chromosome fusion through the same mechanism.

The phylogeny indicates five potential fusion events, defined as closely related genomes including at least one with a fused chromosome, collected from the same household. Four of these

five events are likely independent because they are separated on the phylogeny by other well-supported clades with two chromosomes (Fig. 2C; ultrafast bootstrap approximation support >95). Event 1 is a single fusion event in a household contact, with no evidence of onward transmission (Figs. 2D, S4). In event 5, 7/10 isolates from the index case had a fused chromosome, but 0/10 isolates from the household contact were fused, which appear to be only distantly related to the index case isolates (12–29 SNVs; Fig. S4), again suggesting a single fusion event and no onward transmission. In events 2-4, we observe fused chromosomes amongst the closely related genomes (typically 0–2 SNVs; Fig. S4) isolated from both index cases and household contacts, suggesting transmission of the fusion. The clade containing event 2 includes an index case (with 1 fused chromosome out of 9 genomes sequenced) and contact (5/10 fused) from household P, as well as non-fused chromosomes from household O. This observation is consistent with a variety of scenarios, including a single fusion event within household P, followed by transmission of a mixed fused/non-fused population within the household. Alternatively, only non-fused *V. cholerae* could have been transmitted, followed by independent fusion events in the index and contact within the household P. The clades containing events 3 and 4 each involve a single household with one index and two contacts, all of which contained fused chromosomes (Fig. 2D). The isolates from these two households are closely related and separated by a low-supported branch of non-fused isolates (ultrafast bootstrap approximation support <25). This is consistent with either independent fusion events in each household or a single fusion event stable enough to be transmitted between households. Within each household, these clades are also consistent with transmission of a mixed fused/non-fused population, or with 2–3 independent fusion events. A more parsimonious scenario would be a single fusion event in the index case, followed by fused-chromosome transmission to both contacts, and a single fission event in one contact. Although we cannot formally distinguish between these scenarios, it is clear that chromosome fusion occurs repeatedly and is stable enough to be transmitted within and possibly between households.

### Chromosome fusion is stable under laboratory conditions

To further quantify the stability of chromosome fusion, we conducted a passaging experiment and found the fusion state to be stable over 200 generations under laboratory conditions (Supplementary Data 2, Fig. S5). This apparent stability could be explained if fused isolates are recombination-deficient and therefore unable to reverse the fusion event. However, this does not appear to be the case: genes involved in recombination (*rec* and *mut* genes) did not contain any fusion-specific mutations. To search for other mutations that could potentially stabilize chromosomal fusion, we used a genome-wide association study (Methods) but did not find any protein-coding variants significantly associated with fusion state. Nor did we identify any significant differences in the number of SNVs accumulated during the passage experiment between fused and unfused isolates ($p$-value > 0.05, Mann-Whitney U), which would have been expected had they differed in their recombination or DNA repair proficiency (Fig. S6). In conclusion, we find no evidence that fused isolates are recombination-deficient. Yet, we detected closely-related fused and non-fused isolates collected from the same household and patient, suggesting fusion/fission events can occur within patients, likely on time scales of a few days. How chromosome fusion is stabilized under laboratory conditions and whether fusion/fission events are more dynamic within patients remains to be examined.

### Fused chromosomes identified in publicly available genome sequences

We further investigated the frequency of chromosome fusion more broadly across the order *Vibrionales*, in which a bipartite genome is considered a defining feature. We downloaded publicly available long-read sequences ($n = 302$, 251 of which passed our quality controls), 73.7% (185/251) of which were *V. cholerae* (Fig. S7A). Of the fully circular assemblies ($n = 203$), four assembled to one fused chromosome (Fig. S7B). One of these was a *V. natriegens* genome, whose chromosomes were lab-engineered to be fused[12]. The remaining three were clinical *V. cholerae* isolates[13,14]. Two of these, from the IND1.1 sublineage, contained two directly-oriented copies of HS1, as in our isolates. The third genome, which was related to IND2, also contained two copies of HS1 but in the opposite orientation, suggesting a local inversion of one HS1 copy. In a larger dataset of short-read sequenced *V. cholerae* genomes ($n = 1223$), we observe HS1 mainly to be duplicated in sub-lineage BD2 (Fig. S7C). These results and previous studies[7–9] suggest that fusion, while rarely observed, can occur in different *Vibrio* species and *V. cholerae* sublineages.

### VSP-I could be another potential fusion site

We next asked if HS1 is unique, or if other potential fusion sites exist in the genome. For each circular, non-fused public genome ($n = 199$), we compared chromosome 2 against chromosome 1 for regions of homology. We identified two such regions longer than 10 Kbp in *V. cholerae*, one of which corresponded to HS1 and the other to VSP-I[2] (Fig. S8). Although VSP-I might serve as a potential fusion site, we currently lack evidence for this as none of the fused-chromosome genomes carries more than one VSP-I copy. Whether fusion at VSP-I takes place and is viable therefore remains unknown.

### Defects in chromosome replication are unlikely to explain fusion

Several genes are known to be involved in *V. cholerae* chromosome replication, and these all appear to be present and intact in fused chromosomes. Both *V. cholerae* chromosomes encode two partitioning (*parAB*) genes involved in separating chromosomes to daughter cells[15]. We detect all four *parAB* genes in the fused chromosomes sequenced here ($n = 58$), with no mutations compared to *par* genes on non-fused chromosomes. Fused chromosomes also contain origins of replication from both chromosomes (*ori1* and *ori2*) and *crtS* (*C*hr2 *r*eplication *t*riggering *S*ite). Previous studies showed that the two *V. cholerae* chromosomes fuse temporarily upon deletion of *crtS*[16,17]. *crtS* is located closer to *ori1* than *ori2* is to *ori1* in all fused chromosomes. This arrangement of loci has previously been associated with *ori2* being active in a naturally fused *V. cholerae* chromosome[18]. Finally, chromosome fusion can be selected experimentally through depletion of the DNA adenine methylase Dam[6]. The *dam* gene was present in all genomes described here with no non-synonymous mutations detected; therefore, loss of *dam* function is unlikely to explain the observed fusions. Further experiments will be needed to understand how fused chromosomes replicate and whether fusion affects bacterial growth or other phenotypes under different conditions.

### Conclusions

Together, our results show that chromosome fusion via homologous recombination is more prevalent and potentially more stable than previously thought. The clinical or phenotypic consequences of fusion appear to be minimal but remain to be comprehensively explored. This study reveals that chromosome fusion in clinical *V. cholerae* O1 occurs at an unprecedented scale and highlights the power of long-read sequencing to identify structural variation in bacterial genomes.

## Methods

### Ethical statement

The Ethical and Research Review committees of the ICDDR, B (approval number PR-11041) and the Institutional Review Boards of Massachusetts General Hospital, the University of Washington, and McGill University (A07-M43-21B (21-07-026)) approved the study. All adult subjects in the study provided written informed consent and the parents/guardians of children provided written informed consent.

### *V. cholerae* isolate collection

Stool and rectal swabs were sampled from patients with cholera admitted to the International Centre for Diarrhoeal Disease Research, Bangladesh (ICDDR, B), Dhaka Hospital and from their household contacts, as described in prior studies[19]. Patients presenting to the hospital with severe acute diarrhoea and a stool culture positive for *V. cholerae* O1 were considered index patients. Persons who shared the same cooking pot with the index patient for 3 or more days are considered household contacts and were enrolled within 6 h of the presentation of the index patient to the hospital. Rectal swabs were collected daily from household contacts during a 10-day period after presentation of the index case. Household contacts underwent daily clinical assessment of symptoms. Household contacts were defined as infected if any rectal swab culture was positive for *V. cholerae* O1. *V. cholerae* serotypes were determined using slide agglutination testing with polyvalent and specific antisera as in prior studies[20]. We excluded patients below 2 years of age and above 60 years old or with major comorbid conditions[21,22]. Rectal swabs and stool from the day of enrollment and follow-up time points were collected and placed immediately on ice after collection and stored at −80 °C until DNA extraction.

*V. cholerae* isolates were cultured from stool samples from culture-positive participants. Stool samples were streaked directly onto tellurite taurocholate gelatin agar, a selective medium for *V. cholerae*, and incubated at 37 °C for 18–24 h. Ten colonies from each participant were selected and inoculated into Luria-Bertani (LB) broth (BD Difco) and grown at 37 °C overnight. Liquid cultures were used to make 30% glycerol stocks and stored at −80 °C. Frozen glycerol stocks were then shipped to the University of Washington.

### *V. cholerae* DNA extraction

*V. cholerae* from frozen glycerol stocks were streaked onto LB agar plates and incubated at 30 °C for 24 h. One colony per plate was used for liquid culturing in LB broth and incubated at 30 °C for 18 h with agitation. DNA was extracted from saturated liquid cultures using the DNeasy Blood and Tissue 96-kit (Qiagen) according to the manufacturer's protocol. Briefly, *V. cholerae* samples were pelleted and resuspended in Buffer ATL and treated with proteinase K at 56 °C for 30 min followed by RNase A (Qiagen) at room temperature for 5 min and Buffer AL for an additional incubation at 56 °C for 10 min. Samples were then treated with 100% EtOH and transferred to Qiagen DNA spin columns. DNA was washed using Buffer AW1 and Buffer AW2. Purified DNA was eluted in 100 μL 10 mM Tris-HCl and stored at −80 °C until ready for sequencing.

### DNA sequencing

The extracted DNA was prepared for sequencing using the Nanopore Rapid Barcoding 96 v14 (Oxford Nanopore Technologies) with approximately 200 ng of purified DNA per sample to generate sequencing libraries. The libraries were sequenced on a R10.4.1 M PromethION flow cell. Raw sequencing data was basecalled and demultiplexed to FASTQ files using the Dorado basecaller integrated into MinKNOW v 23.07.5 (Oxford Nanopore Technologies) using the model dna_r10.4.1_e8.2_400bps_sup@v4.2.0 with read splitting, adapter trimming, and barcode trimming enabled in the basecaller.

Read data generated for this study is available on NCBI SRA (BioProject PRJNA1121190).

### Sequence analysis of isolates collected for this study

**Assembly, quality control and annotation.** Reads of each sample were assembled using Flye[23,24] in –nano-hq and deterministic mode. We used dnaapler[25] to reorient the contigs such that all chromosomes start with the initiating codon of *dnaA* if present (chromosomes 1 and fused chromosomes) or with *repA* homologous sequences if *dnaA* was not present (chromosomes 2). We used ReCycled[26] to verify the circularity for all contigs. We used Kraken2[27] to identify potentially contaminated assemblies and screened these using SprayNPray[28], which assigns species membership per contig. All contigs that were not identified as *V. cholerae* were removed from the assemblies. These included 50 short contigs with repetitive sequences and no species membership assigned, likely arising from sequencing artifacts.

As a control, we sequenced and assembled the reference strain *V. cholerae* N16961, which assembled into two chromosomes, as expected. To further reduce the chances of misassembly, we manually assembled sequences of 13 isolates using Trycycler[29]. This subset included one putative fused-chromosome isolate per patient for which at least one fused-chromosome was assembled (n = 10) and three non-fused chromosome isolates as references. We followed the Trycycler workflow. First, the reads of each sample were filtered using Filtlong[30] (minimum length 1000 bp and excluding the worst 5% of read bases) and subsampled to 12 files. Of each isolate, 3 subsets were assembled with Flye[23], Miniasm[31] and Minipolish[32], Raven[32,33] and Canu[32–34]. All assembled contigs were clustered per sample, aligned, and manually curated. We considered contig clusters as valid if they (i) occurred in at least two assemblies and were of similar size, (ii) no kilobase of sequence was below the identity threshold of 25% when compared to the rest of the contig, and (iii) could be circularly assembled. Of each valid contig cluster, a consensus was built, which together form the assembly for each sample. The assemblies resulting from the Trycycler workflow confirmed the chromosome fusion state observed from the Flye assemblies. We used the Flye assemblies for all isolates for all subsequent analysis. We used Nanostat[35] to assess the quality of our assemblies. All assemblies were annotated using Bakta[36,37].

**Phylogeny.** We called variants of each of our isolates to the *V. cholerae* reference genome N16961 (GCF_001250235.2) using Medaka[38] with a minimal quality score threshold of 40 and excluding indels using vcftools[39]. We identified varying sites from the resulting consensus sequences using snp-sites[40] and constructed a maximum-likelihood phylogenetic tree using IQtree (v2.3.6)[41] and assessed branch support using UltraFast bootstrapping approximation[42].

**Sub-lineage assignment.** We aimed to assign our isolates to one of the sub-lineages within the clonal lineage causing the current 7th pandemic (7PET). We selected one previously assigned and publicly available genome per sub-lineage[5] as a reference and called variants of each of our sequences to each reference using the Medaka variant caller[38]. Each isolate was assigned to the sub-lineage to whose reference the smallest number of high-quality variants was identified (using a minimal quality score threshold of 40).

**Identification of HS1.** To identify potential fusion sites, we used blastn[43], identifying sequences that are shared between chromosome 1 and chromosome 2. We detected a 12 Kbp-long sequence that was identical between chromosome 1 and chromosome 2 in most strains (HS1) (>99.6% sequence identity in assemblies with two copies of HS1). We screened for the occurrence of HS1 in all our genomes using blastn[43].

To further substantiate the observed chromosome fusion, we next aimed to identify reads that span HS1 and are either flanked by

chromosome 1 or chromosome 2 sequences (non-fused case) or are flanked by sequences of each chromosome on each side (fused case). To do so, we mapped the reads of each sample to a fused-chromosome assembly (135Vc06) and a non-fused-chromosome assembly (135Vc05). From the bam file, we extracted reads that span the HS1 and an additional 500 additional bp on each side using samtools. We excluded secondary mappings and mappings with large insertions (>1000 bp).

We excluded reads in which a sequencing adapter sequence was identified, as these could potentially be artifactual hybrid molecules. To do so, we converted the reads to FASTA files and screened for adapter sequences using blastn. All reads in which an adapter sequence with >95% coverage and identity was identified were excluded from further analysis.

**Screening of genomic islands.** We queried our genomes for the presence and location of the virulence-associated genomic islands VPI-1, VPI-2, VSP-I, and VSP-II. We screened our genomes for the known flanking regions[44] of these islands using blastn[43] and extracted the sequence between the flanking sides using bedtools getfasta[44,45].

### Comparison of sequences involved in chromosome replication and recombination

**par genes.** *V. cholerae* typically encodes two sets of partitioning (*par*) genes (*parAB* on chromosome 1 and *parAB2* on chromosome 2), which are involved in separating the chromosome molecules to daughter cells. As chromosome 2 can be lost upon depletion of *parAB2*, we hypothesised that *parAB2* might be lost or mutated in strains with a fused chromosome. To test this, we downloaded the amino acid sequence of reference *V. cholerae par* proteins (QEO41389.1, QEO42567.1, AAF97006.1, and AAF97005.1 for ParA, ParB, ParA2, and ParB2, respectively) and screened their occurrence in our genomes using tblastn[43].

**dam and crtS.** To compare the *dam* gene sequence and the presence of the origins of replication on the fused chromosomes, we examined the Bakta annotation. We screened for *crtS* (*C*hr2 *r*eplication *t*riggering *S*ite)[18] in our genomes using blastn[43].

**mut and rec genes.** To infer the phylogeny, we had called variants of each of our isolates to the *V. cholerae* reference genome N16961 (GCF_001250235.2) using Medaka[38] with a minimal quality score threshold of 40 (see above) and annotated these using snpEff[46]. We then evaluated in R whether any SNVs were identified within the N16961 *mut* or *rec* genes. *V. cholerae* N16961 was annotated using Bakta.

### Comparison to previously reported chromosome fusion

We extracted sequences described by Xie et al.[9] to be either directly involved in homologous recombination (VAA049_1594, VAA049_2432 from NCSV1 and VAB027_307, VAB027_1228 from NCSV2) as well as the IS3 elements (VAA049_2433 from NCSV1 and VAB027_276, VAB027_1254 from NCSV2) as amino acid sequences from the published fused chromosome genomes (NCSV1 (NZ_CP010811.1) and NCSV2 (NZ_CP010812.1)). We compared these to HS1 using tblastn and found no matches with a sequence identity >60% or a query coverage >37%, suggesting that these are not the same sequences.

### Genome-wide association study

Aiming to explain the stability of the fused chromosome under laboratory conditions, we used pyseer[47] (v1.3.12) to identify sequence variants associated with fused-chromosome genomes. We used unitigs (non-redundant sequence elements of variable lengths) as inputs, which we constructed via unitig-counter[48] (v1.1.0). We used random effects to correct for population structure ('lmm' mode) using the phylogenetic tree as input (compiled from high-quality core SNV, see above). The number of unique patterns ($n = 1036$) was used to determine a significance threshold ($p$-value $< 4.83E{-}05$). Significant unitigs were mapped against the Bakta genome annotations from all genomes ($n = 467$).

### *V. cholerae* plug preparation and pulsed-field gel electrophoresis (PFGE)

*V. cholerae* isolates from frozen glycerol stocks were cultured as described above. *V. cholerae* plugs were created as previously described (CDC Pulsenet protocol)[49]. Briefly, *V. cholerae* cells were inoculated into a cell suspension buffer (100 mM Tris, 100 mM EDTA, pH 8.0) to an optical density (OD) of 2.0. Plugs were cast using 0.5 mg/mLProteinase K (ThermoFisher) and 1% SeaKem Gold agarose (Lonza) prepared in Tris-EDTA (TE) buffer (10 mM Tris, 1 mM EDTA, pH 8.0). Plugs were lysed in cell lysis buffer (50 mM Tris, 50 mM EDTA, pH 8.0 with 1% Sarcosine (Sigma) and 0.1 mg/mL Proteinase K), incubated at 55 °C for 30 min with shaking at 200 rpm. Plugs were then washed twice with sterile, distilled water and four times with TE buffer. Plugs were stored in TE buffer at 4 °C until PFGE was performed.

Gels, 21 cm long, were cast for PFGE using 0.8% SeaKem Agarose in 1X Tris acetate EDTA (TAE) buffer. PFGE was performed on a CHEF-DR III system (Biorad) with runtime of 60 h, at 2 Volts/cm with 30 min switch time (initial and final) at a 106° angle with 1X TAE running buffer chilled to 14 °C. The gel was then stained using SYBR Gold (Invitrogen) and visualised using Amersham Typhoon 5 (Cytiva). *Schizosaccharomyces pombe*[50] DNA was used as a DNA marker.

### Phenotypic comparison of fused and non-fused isolates

**Isolate selection.** To assess phenotypic effects of chromosome fusion, we chose two pairs of fused and non-fused isolates, each collected from the same participants (114 and 117) and with a minimal number of SNVs between isolate pairs. SNVs were called using clair3 using the non-fused strains 114Vc01 and 117Vc10 as reference and 114Vc03 and 117Vc01 as query, and a minimal quality score of 20, minimal read depth of 10 and minimal ALT allele frequency of 0.8.

### qPCR measurements of *ctxA* and *tcpA*

**V. cholerae virulence-inducing conditions.** Liquid cultures of *V. cholerae* were obtained in LB broth, 37 °C with agitation. Saturated cultures were then diluted to OD = 0.01 into 10 mL AKI media containing 1.5% peptone (HIMEDIA), 0.5% sodium chloride (Fisher Bioreagents), 0.4% yeast extract (Fisher), and 0.3% sodium bicarbonate (Fisher Chemical). Diluted cultures were incubated stationary at 37 °C for 4 h and then switched into a 250 mL flask with shaking at 250 rpm for 3 h. After the incubations, the cultures were centrifuged at 4,000 x g for 10 min at 4 °C. Pellets were resuspended in 1 mL TRIzol (Invitrogen) and stored at 4 °C for RNA extraction.

**RNA extraction and Real-time PCR (RT-PCR).** Samples in TRIzol had RNA extracted using chloroform (Fisher Chemical), ethanol (Fisher Chemical), and RNeasy kit (QIAGEN) according to the manufacturer's protocol. Extracted RNA was treated with TURBO DNase (Invitrogen) to remove DNA. cDNA synthesis was performed on 1 µg of RNA using High Capacity cDNA reverse transcriptase (Applied Biosystems) and resulting cDNA was diluted 1:3. RT-PCR was performed to measure *ctxA* and *tcpA* expression. The primers used were (CtxA-F) 5′-TTGGAG-CATTCCCACAACCC-3′, (CtxA-R) 5′-GCTCCAGCAGCAGATGGTTA-3″ – amplicon 109 bp[51], (TcpA-F) 5′-CGCTGAGACCACACCCATA-3′, (TcpA-R) 5′-GAAGAAGTTTGTAAAAGAAGAACACG-3′ – amplicon 103 bp[52], (groEL-F) 5′-ATGATGTTGCCCACGCTAGA-3′, and (groEL-R) 5′-GGTTATCGCTGCGGTAGAAG-3′ – amplicon 117 bp[52]. GroEL was used as a reference housekeeping gene. RT-PCR was performed using a 10uL reaction with SYBR green (Invitrogen) with 0.3 µM of specific primer sets and 2 µL of cDNA. PCR amplification was conducted on the

QuantStudio3 (ThermoFisher Scientific) with the following conditions: 95 °C for 5 min, 40 cycles of 95 °C for 5 s, 58 °C for 10 s, and 72 °C for 15 s, and a final melting temperature analysis of PCR products. Each RT-PCR run included a no-template and water negative control. Each sample was tested in duplicate. Relative expression was calculated according to the Livak method[53].

## Biofilm crystal violet assay

Liquid cultures of *V. cholerae* were generated from single colonies in LB broth at 37 °C with agitation. Saturated cultures were normalized to OD = 0.01 in 200 µl into 96-well flat-bottom clear polystyrene microplates, sealed with a gas-permeable sealing film (BrandTech). Plates were incubated at 30 °C for 24 h. Crystal violet staining was performed as previously described[11,54]. Briefly, bacterial cultures were removed, and plates were washed with distilled water by submerging the plates three times. Adherent biofilm was stained using 0.1% crystal violet (Fisher Chemical) at room temperature for 15 min. Excess crystal violet was removed, and the plate was air-dried for 3 h. The Crystal violet stain was then dissolved in 95% ethanol for 15 min, and the absorbance was measured at 550 nm using a Synergy HT plate reader (Biotek) with Gen5 software (Agilent BioTek).

## Growth curves

**Growth curves with 96-well plate**. Liquid cultures of *V. cholerae* were obtained in LB broth at 37 °C with agitation. Cultures were then diluted to OD = 0.01 in 200 µl into 96-well flat-bottom clear polystyrene microplates, sealed with a gas-permeable sealing film (BrandTech). Plates were incubated at 37 °C for 20 h in the Stratus plate reader (Cerillo) with continuous OD600 measurements every 20 min.

**Growth curves in roller drum**. Liquid cultures of *V. cholerae* were obtained in LB broth at 37 °C with agitation. Cultures were then diluted to OD = 0.01 in 10 mL LB broth and incubated in a roller drum at 37 °C for 24 h. OD600 was measured using a 1 mL in a cuvette (Fisherbrand) in a Cell Density Meter (Transilluminator) at 0, 2, 4, 6, 8, 10, and 24 h.

## Continued passaging of *V. cholerae*

**Experimental setup**. *V. cholerae* was continuously passaged as previously described[18]. Briefly, *V. cholerae* isolates were plated on LB agar and incubated at 30 °C for 72 h. On Day 1, one CFU was inoculated into 10 mL LB broth in a 15 mL snap-cap polystyrene tube and incubated at 37 °C for 24 h in a roller drum. The following day, saturated cultures were diluted 1:10000 into fresh LB broth, and this procedure was repeated daily. On Day 5, liquid cultures were streaked on LB agar and incubated at 30 °C for 72 h. This process was repeated for 20 days. Glycerol stocks using 20% glycerol (Fisher Bioreagents) in LB broth were periodically saved at −80 °C, and liquid cultures were streaked on agar to check for contamination twice a week.

## Sequence analysis

Single colonies were obtained from the glycerol stocks collected on day 0, 3, 8, 11, 16, and 20 of the passaging experiment. *V. cholerae* isolates were cultured in LB broth and DNA was extracted as described above. DNA was prepared for long-read whole-genome sequencing and sequenced on a R10.4.1 M PromethION as described above. Genomes with an estimated read coverage of less than 10X were excluded from further analysis (Supplementary Data 2).

Reads of each sample were assembled using Flye[23] in –nano-hq and deterministic mode. We evaluated how many chromosomes were assembled and counted reads that mapped HS1 in its fused-chromosome environment (chromosome 1-HS1-chromosome 2 and chromosome 2-HS1-chromosome 1) and in its non-fused environment (chromosome 1-HS1-chromosome 1 and chromosome 2-HS1-chromosome 2) (see 'Identification of HS1').

We called SNV of each isolate compared to its source isolate (114Vc01, 114Vc03, 117Vc01, and 117Vc10) using clair3[55] and minimal read depth >=10, minimal allele frequency >=0.8 and minimal quality >=40. We used snpEff[46] to annotate the identified SNV.

## Sequence analysis of publicly available sequences

**Selection of publicly available sequences**. To contextualise the sequences acquired for this study, we used two different datasets, which we compiled from publicly available sequences: (a) long-read sequenced genomes of the order *Vibrionales* and (b) short-read sequenced genomes of the species *V. cholerae*.

Dataset (a) consisted of all *Virbrionales* genomic sequences that were acquired using Oxford Nanopore Technology (ONT) or Pacbio and available from NCBI SRA on the 19th of December 2023 (*n* = 302).

We compiled dataset (b), aiming to reflect the diversity within clinical *V. cholerae* and to focus our study period (2015-2018) and geographic region of isolation (Bangladesh) for comparison. To do so, we downloaded the following sets of short reads: (b.i) Clinical *V. cholerae* overview: from NCBI pathogens, we selected *V. cholerae* which were collected between the 1st of January 2000 and the 18th of January 2024 and included a maximum of three samples per year and country (*n* = 790, 457 of which could be successfully downloaded using NCBI Batch entrez).

(b.ii) Diversity within *V. cholerae* O1: We downloaded a previously compiled selection of *V. cholerae* O1[56] (all O1 isolates from this study).

(b.iii) 7PET diversity from South and East Asia: Of a previously compiled 7PET set of genomes[57], we included strains which were isolated between 2000 and 2018 in Eastern Asia, Southeast Asia, and South Asia. Moreover, we included the sequences acquired in this[57] and another study conducted in Bangladesh around our study period[58].

**Assembly, quality control and annotation**. Long-read sequences were assembled using Flye[23,24] (--nano-raw mode) and the species was identified using GTDB-tk[59]. We controlled the quality of the assemblies using Kraken2[27] / Bracken[60], as well as Nanostat[35]. We excluded sequences which were not identified as part of the order *Vibrionales*, where >10% of the contigs were not assigned as the most abundant genus (suggesting contamination), whose total assembly length was smaller than 3.5 or larger than 7 MB, or which had an average read depth below 5X, resulting in 251 genomes of 24 different species.

Short-reads were trimmed using Trimmomatic[61] and were assembled using Spades[62] via Unicycler[63]. We assessed the quality of the resulting assemblies using Quast[64] and used Metaphlan2[36] to screen for possible contaminations. We excluded sequences that had a Metaphlan purity below 95%, an average read depth below 20X, an average read quality below 75, where the resulting assembly was shorter than 3.65 or longer than 4.5 MB, or had N50 value below 2000. This resulted in a set of high-quality *V. cholerae* short-read genomes (*n* = 1,223). The accession numbers of all publicly available sequences analysed in this study can be found in Supplementary Data 3.

## Phylogeny

For the *Vibrionales* long reads, we built an SNV-based phylogeny, as described above. Briefly, we called variants of each sample to the *V. cholerae* reference genome N16961 (GCF_001250235.2) using Medaka[38] (minimal quality threshold 40), identified varying sites of the resulting consensus sequence using snp-sites[40], and built a phylogenetic tree using RaxML (v8.2.12, model GTRCAT)[65].

We built a core genome for the *V. cholerae* short reads using panaroo[66] and aligned it using mafft[67]. Variant sites were identified using snp-sites[40], and a phylogenetic tree was constructed using fasttree[68].

## (Sub)-Lineage assignments

We used 'Is it 7PET'[69] to identify which publicly available *V. cholerae* sequences belonged to the 7PET lineage. To identify the sub-lineages within the 7PET, we quantified the number of SNV to one reference per sub-lineage each, and assigned the sub-lineage with the least number of SNV, similar as described above. SNV were called using Freebayes[70] via snippy[71] (minimum read depth = 10, minimum fraction = 0.7, and minimum quality = 100).

## Copy number estimation of potential fusion sites

We used blastn[43] to identify the presence and location of the potential fusion sites HS1 and VSP-I in the publicly available *V. cholerae* genomes (>95% identity and >90% coverage). We used bwa[72] to align the reads of each isolate to its respective assembly. From the resulting bam files, we retrieved the average read depth of the genome and from the specific potential fusion sites using samtools depth[73].

## Reporting summary

Further information on research design is available in the Nature Portfolio Reporting Summary linked to this article.

## Data availability

Sequence data generated for this study (raw reads and assemblies) have been deposited on NCBI under the accession code PRJNA1121190. All sequence analysis data generated, which are required to reproduce the figures and analysis presented in this study, are provided in the Source Data file and via the OpenScienceFoundation (https://osf.io/xyfvg/). Source data are provided with this paper.

## Code availability

Software code used to analyse the data presented in this study is available on GitHUb (https://github.com/acuenod111/Single_chromosome_Vc; https://doi.org/10.5281/zenodo.15298067) and the files to reproduce the analysis and figures can be accessed via the OpenScienceFoundation (https://osf.io/xyfvg/).

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

## Acknowledgements

We are grateful to the people of Dhaka, where our study was undertaken, to the field, laboratory, and data management staff, who provided a tremendous effort to make the study successful, and to the people who provided valuable support in our study. We also acknowledge Matthew Doucette for sharing protocols. AC was supported by a Postdoc. Mobility Fellowship from the Swiss National Science Foundation (P500PB_214356). BJS and PL were supported by a Canadian Institutes for Health Research (CIHR) Project Grant and Fellowship, respectively. This work was also supported by a grant from the U.S. National Institutes of Health/NIAID AI106878 (ETR, FQ), K08AI123494 (AAW), and T32HD007233 (DC). RWH and GRS were supported by research grant R35 GM118120 from the National Institutes of Health of the United States of America.

## Author contributions

A.I.K., F.C., S.B.C., E.T.R., J.B.H., R.C.L., T.R.B., and F.Q. conducted the study in Dhaka, Bangladesh, and collected the patient samples. D.C. extracted the DNA for all samples. D.C., A.R. conducted the in vitro experiments. P.L. developed the sequencing workflow and compiled the initial assemblies. P.L. and A.K. sequenced all samples. A.C. conducted all other bioinformatic analysis and generated the figures. D.C., R.W.H., G.R.S., and S.M.M. designed and performed the PFGE assay. A.A.W., B.J.S., and P.L supervised the project. E.T.R., F.Q., D.C., G.R.S., B.J.S., A.A.W., P.L., and A.C. acquired funding for this project. A.C. and B.J.S. wrote the original manuscript. D.C., A.I.K., F.C., R.W.H., S.M.M., A.R., A.K., S.B.C., E.T.R., J.B.H., R.C.L., T.R.B., G.R.S., F.Q., P.L., and A.A.W. reviewed and approved the manuscript.

## Competing interests

The authors declare no competing interests.
