## [Transparent Peer Review file · Nature Communications]

Prevalent chromosome fusion in *Vibrio cholerae* O1

Corresponding Author: Dr B. Jesse Shapiro

Version 0:

Reviewer comments:

Reviewer #1

(Remarks to the Author)

Review for the Manuscript : Prevalent chromosome fusion in *Vibrio cholerae* O1

The genome of *Vibrio cholerae* (Vc), like other members of the Vibrionaceae family, is typically bipartite, consisting of two circular chromosomes: the larger ~3 Mbp chromosome 1 (Chr1) and the smaller ~1 Mbp chromosome 2 (Chr2). This bipartite structure is a defining characteristic of the genus. Chr1 contains most housekeeping genes, whereas Chr2 carries mostly genes that contribute to the organism's adaptability and survival in different environments, including virulence factors.

However, there have been rare reports of naturally occurring Vc isolates with a single fused chromosome. These fusions are typically thought to occur via homologous recombination at specific repetitive sequences shared between the two chromosomes. The study under review challenges the notion that such fusion events are rare.

The authors have investigated chromosome fusion in Vc using long-read nanopore sequencing of 467 clinical isolates collected from cholera patients in Dhaka, Bangladesh between 2015 and 2018. The study identifies fused chromosomes in 58 isolates. In all cases, fusion occurred via homologous recombination between a 12 Kbp homologous sequence (HS1) present on both chromosomes. This work provides new insights into the prevalence and stability of chromosomal fusions in clinical isolates of Vc. It reveals that chromosome fusion is more prevalent and potentially stable than previously thought. This study also highlights the power of long-read sequencing in identifying structural variations in bacterial genomes.

Major points

1. Improve the discussion of phenotypic consequences and potential selective advantage :

By demonstrating a higher prevalence of these fusions in clinical isolates, the authors provide new insights into the genomic plasticity and adaptive strategies of Vc, with implications for understanding pathogenesis, and potential impacts on bacterial fitness and virulence. The manuscript could delve deeper into discussing the potential phenotypic consequences of chromosome fusion, such as impacts on virulence or other relevant bacterial traits.

2. Add context from other studies of spontaneous chromosome fusions in *Vibrio* :

The discussion could benefit from a more comprehensive comparison with previous studies on natural chromosomal fusions in *Vibrio*, especially as the mechanism of fusion by homologous recombination could be similar. For instance:

Xie et al. (2017) (doi: 10.1155/2017/8724304) analyzed two naturally fused strains, NSCV1 and NSCV2, and identified probable homologous recombination regions that could be related to those in this study (IS3 related).

Val et al. (2014) (doi:10.1111/mmi.12483) investigated potential fusion sites and observed that all homologous recombination-mediated fusion sites occurred between ISVch4 elements of the IS3 family, also found in HS1 in this study.

3. Line 142: What is crtS and why is its location important?

Refer to the study by Val et al. (2016) (doi: 10.1126/sciadv.1501914) where the importance of crtS location in the timing of Chr2 replication is shown. When crtS is deleted, chromosome fusion occurs spontaneously as an intermediate before fission (see point 4).

Figure 2A. The schema of Chr1-Chr2 fusion is rather too simple. Locate at least the origins of replication, the crtS site and some landmark regions (e.g. virulence-associated genomic islands VPI-1, VPI-2, VSP-I and VSP-II).

4. Stability of fused chromosomes:

The authors suggest that fused chromosomes can remain stable for ~12-22 years based on substitution rates. However, rapid fusion/fission events observed within the same household or patient indicate a dynamic process needing further investigation.

Not being an expert in phylogeny, I can't assess the accuracy of the phylogenetic tree or the statistical calculations used to estimate the stability period.

- The study could clarify whether the fission events resolved from the same HS1 element or whether other mechanisms are involved that could promote genetic exchange between the two chromosomes.

- Chromosome fusions have previously been shown to be transient. In Val et al. (2016), using crtS mutants, chromosome fusions revert to a two-chromosome configuration over 200 generations after acquiring compensatory mutations. This suggests that fusions are generally not stable without specific selective pressures. The author could test the capacity of some of their fused strains (e.g. 111Vc05, 112Vc01...) to revert to two chromosomes under lab conditions. Some of these strains might be blocked in the fused states due to mutations that prevent reversion. Passaging these strains for 200 generations and performing PFGE should suffice to explore this possibility. This experiment would strengthen the assertions made in the manuscript.

5. Line 130-135: VSP-I as a potential fusion site:

While VSP-I was identified as a potential homologous sequence for fusion, no evidence was found to support its involvement in the chromosome fusions observed. The manuscript could benefit from discussing why VSP-I might not be used or selected. Would the resulting fusion be non-viable, or viable but not advantageous, and therefore not selected?

Minor points

Figure 1C – PFGE : What is the smaller band observed in all samples?

Line 145: In this study, Vc was not “engineered.” The chromosomes fused spontaneously and were selection upon dam depletion.

Reviewer #2

(Remarks to the Author)

Here the authors report evidence of chromosome fusion in *Vibrio cholerae* isolated from clinical cases in Bangladesh. In several of the households studied, chromosome fusion was detected in both cases and contacts, suggesting that *V. cholerae* with a single chromosome can be transmitted. Results were validated using manual assembly and pulsed-field gel electrophoresis to ensure the results were not an artefact. These results are highly novel and will have significant impact on our understanding of the biology of *Vibrio cholerae*. It is not yet clear what the clinical relevance of such a finding is and this will be an interesting avenue of further research.

Line 53: Is there any data to investigate whether there is an association between fusion and a clinical phenotype, or a participant characteristic such as occupation?

Line 69: If there is a heterogenous population, some with fusion and some without, why would there not be a band at 3Mbp?

Line 84/Figure 2A: Would it be possible to indicate on the schematic where HS1 is in relation to the origins of replication? Does the fusion bring VPI-2 closer to an origin of replication, and could this affect its expression?

Line 86/Figure 2B: Adding the corresponding locus tags (e.g. VC1757) for the genes in VPI-2 and chromosome 2 would make it clearer to readers where the homologous sequences are in the genome. What is the genomic context of HS1 in chromosome 2?

Line 95/Figure 2C: It would make the figure clearer to have fusion events inferred by ancestral reconstruction shown in Figure 2C, for example by coloring the branches.

Line 100: By ancestral reconstruction the authors detect 10 fusion and 17 fission events. However as samples are taken from household studies, many pairs of samples will be extremely closely related if not identical. The trees shown in Figure 2 and S3 are both bifurcating, and in Figure 2 branch lengths of length zero are illustrated with a small minimum branch length while S3 is a cladogram. It is therefore not possible to tell from these figures which samples are identical. Could the high number of fusion and fission events in the tree be an artefact of this? Within a household, what is the SNP distance between fused and two-chromosome samples?

Line 103: What does subsequent mean in this context? Is it the relationship between a node inferred to represent a fission event, and a parent node also inferred to represent a fission event? Or does it refer to the time that samples were collected (e.g. if you were to detect a fission event in a household on day 2, then another on day 9)? Would you not need to calculate the distance between a fusion and subsequent fission event to estimate the stability of the fusion, rather than between two fission events?

Line 132: Assuming there is no duplication, would I be right in thinking both two-chromosome and fused-chromosome *V. cholerae* contain one copy of HS1 in VPI-2, one copy of HS1 in chromosome 2, and one homologous region in VSP-I? If so, why would the lack of VSP-I copies in the genome preclude it as a recombination site?

Table S1: What unit is median read length in? Please add accession numbers.

Reviewer #3

(Remarks to the Author)

In this article titled "Prevalent chromosome fusion in *Vibrio cholerae* O1" Cuenod et al report (ad verbatim) "chromosome fusions in clinical *V. cholerae* O1 isolates, including several independent fusion events stable enough to be transmitted between patients within a household. Fusion occurs in a 12 kilobase-pair homologous sequence shared between the two chromosomes, which may lead to reversible chromosomal fusion". The authors claim, "together, these results show that chromosome fusion via homologous recombination is more prevalent and potentially more stable than previously thought. The clinical or phenotypic consequences of fusion, if any, remain to be explored. This study reveals chromosome fusion in clinical *V. cholerae* O1 strains at an unprecedented scale and highlights the power of long read sequencing to identify structural variation in bacterial genomes."

While it is true and known that long read sequencing is useful in revealing these types of large structural variations, the other claims of this manuscript are not fully supported. As the authors noted, single chromosome *Vibrio cholerae* (SCV) have been reported in clinical and non-clinical environmental isolates. In fact, similar mechanisms leading to chromosome fusion via homologous recombination have been described in this paper (<https://doi.org/10.1155/2017/8724304>). Did the authors check if they are the same elements or locations involved in those fusion events to rule out that the events described here (12 kb homology in all 58 SCV isolates) are different or different from the Yamamoto et al strain sequence (reference 8)? This leads this reviewer to wonder what is unique in this study.

The authors claim the fusion events are much more prevalent and stable enough to be transmitted within households. This reviewer is not convinced that these are independent events. How do the authors conclude that these are independent events (all occurring at the same location by the same homology)? Is this conclusion based on phylogenetic diversity of fusion strains? Do they not see the same level of diversity in unfused strains? What is more critical for the prevalence argument is this: 58 out of 467 or 458 (12%) to be precise, may appear to be a high proportion compared to the 2 out of 91 (2.1%) fusion events found in naturally occurring environmental non O1/non O139 strains (<https://doi.org/10.1371/journal.pone.0120311>). In that study, they were shown to be independent events (fusions locations and elements are different). Hence, it is possible that these 58 are probably clonal expansion from single event rather than independent events. This hypothesis (clonal expansion) makes sense given the close spatio-temporal proximity and association of the patients and isolates. Perhaps, *sensu lato*, I would argue that there may have been '5 events' in 5 households and those clonal isolates got transmitted between patients in that household. On the opposite spectrum, *sensu stricto*, it might have been a single rare event in the whole site (village) where the samples were collected. Compared to other outbreaks, the SCV was probably prevalent in this outbreak and there was a mix of two chromosome *Vibrio* (TCV) and SCVs causing the outbreak. Unless the clonality issue is unequivocally resolved, (in TCV vs SCV isolates), the claim that all 58 are independent events is not substantiated. Figure 2 somewhat addressed this question. It is not clear where the single chromosome strains are in this tree? Do they cluster or dispersed throughout the tree? The heat map indicated most SCV isolates were from 2 households (green and purple on top). It would really help if a simple table capturing all the isolate metadata, as shown in the below example with these headers, is presented.

Household # (H) Patient # (P) Type [(index I/Sec (S))] Total sequenced Fused Bipartite Colony # H/P/T

The last 2 columns info for all 58 SCV colonies. This table would inform the reader on whether there is bias and clonal expansion or not.

The second major criticism is this: What is not clear is that how the fusions are stably maintained without 'fission'. That is, how are they locked in the SCV configuration? I would expect an equilibrium of fused and separated chromosomes in a population of bacteria *in vivo* in patients as well as *in vitro* (in cultures grown for gDNA extraction) unless the SCV is recombination deficient after fusion. Did the authors check if the bacteria have intact *recA*, for example, or if the bacteria are recombination proficient by other phenotypes; for example, UV resistance or spontaneous mutation frequencies resulting from homologous recombination? Both are easy phenotypes to check. The authors may argue that these are not focus of the manuscript. But these are critical questions that need to be addressed to make this a unique study with significant new findings.

There are additional minor comments throughout the manuscript that might improve/clarify certain part of the manuscript. Please see the edited version.

Version 1:

Reviewer comments:

Reviewer #1

(Remarks to the Author)

The authors have addressed all my concerns in the revision. The additional experiments, analyses, and edits have strengthened the manuscript, and I have no further comments.

(Remarks on code availability)

Reviewer #2

(Remarks to the Author)

I'd like to thank the authors for their responses to the comments. The clarity of Figure 2 has been greatly improved, and even though the phenotypic experiments did not show any major differences I think this adds a lot to the paper.

(Remarks on code availability)

Reviewer #3

(Remarks to the Author)

The authors have addressed the comments, criticisms and suggestions raised by this reviewer. No further questions. The manuscript is ready for the next step.

(Remarks on code availability)

REVIEWER COMMENTS

Reviewer #1 (Remarks to the Author):

Review for the Manuscript : Prevalent chromosome fusion in *Vibrio cholerae* O1

The genome of *Vibrio cholerae* (Vc), like other members of the Vibrionaceae family, is typically bipartite, consisting of two circular chromosomes: the larger ~3 Mbp chromosome 1 (Chr1) and the smaller ~1 Mbp chromosome 2 (Chr2). This bipartite structure is a defining characteristic of the genus. Chr1 contains most housekeeping genes, whereas Chr2 carries mostly genes that contribute to the organism's adaptability and survival in different environments, including virulence factors.

However, there have been rare reports of naturally occurring Vc isolates with a single fused chromosome. These fusions are typically thought to occur via homologous recombination at specific repetitive sequences shared between the two chromosomes. The study under review challenges the notion that such fusion events are rare.

The authors have investigated chromosome fusion in Vc using long-read nanopore sequencing of 467 clinical isolates collected from cholera patients in Dhaka, Bangladesh between 2015 and 2018. The study identifies fused chromosomes in 58 isolates. In all cases, fusion occurred via homologous recombination between a 12 Kbp homologous sequence (HS1) present on both chromosomes. This work provides new insights into the prevalence and stability of chromosomal fusions in clinical isolates of Vc. It reveals that chromosome fusion is more prevalent and potentially stable than previously thought. This study also highlights the power of long-read sequencing in identifying structural variations in bacterial genomes.

Response:

We thank the reviewer for the clear summary of our work and for their appreciation of our key findings.

Major points

1. Improve the discussion of phenotypic consequences and potential selective advantage :

By demonstrating a higher prevalence of these fusions in clinical isolates, the authors provide new insights into the genomic plasticity and adaptive strategies of Vc, with implications for understanding pathogenesis, and potential impacts on bacterial fitness and virulence. The manuscript could delve deeper into discussing the potential phenotypic consequences of chromosome fusion, such as impacts on virulence or other relevant bacterial traits.

Response:

To address these comments we performed growth comparisons between two pairs of closely related fused (one chromosome) vs. non-fused (two chromosome) isolates (n=2 pairs). We found no significant differences, and have added this data to Fig. S2A-B. To assess differences in virulence, we measured cholera toxin and toxin co-regulated pilus gene expression, the two most important and necessary virulence factors for human infection. We

measured these by qPCR and also measured *V. cholerae* biofilm formation in the fused and non-fused isolate pair, and this data is shown Fig. S2C-D. Again, there were no detectable differences between fused and non-fused isolates. We have added the following to the main text:

“Comparing closely related fused and non-fused isolates from the same patient (n=2 pairs, 0 and 2 high quality SNV) we found no difference in growth, or in expression of the key virulence factors cholera toxin (*ctx*) and toxin co-regulated pilus (*tcp*), or in their capability to form biofilms (Fig. S2). This does not exclude the possibility of subtle phenotypic differences that could be detected with larger sample sizes or with other assays.”

2. Add context from other studies of spontaneous chromosome fusions in *Vibrio* :

The discussion could benefit from a more comprehensive comparison with previous studies on natural chromosomal fusions in *Vibrio*, especially as the mechanism of fusion by homologous recombination could be similar. For instance:

Xie et al. (2017) (doi: 10.1155/2017/8724304) analyzed two naturally fused strains, NSCV1 and NSCV2, and identified probable homologous recombination regions that could be related to those in this study (IS3 related).

Val et al. (2014) (doi:10.1111/mmi.12483) investigated potential fusion sites and observed that all homologous recombination-mediated fusion sites occurred between ISVch4 elements of the IS3 family, also found in HS1 in this study.^{1,2}

Response:

We compared the HS1 sequence (described in our manuscript) to the previously published recombination sequence by Xie et al¹, Val et al², and Yamamoto et al^{2,3}. We found that indeed the IS3 elements on HS1 are the same as described by Val et al. and Yamamoto et al. To clarify this we added the VC locus tags to Fig. 2B and added the following to the main text of this manuscript:

“On chromosome 1, HS1 is located within VPI-2 and contains genes VC1788-VC1803 whereas on chromosome 2 HS1 was inserted between VCA0451 and VCA0453. HS1 encodes multiple proteins linked to horizontal gene transfer (Fig 2B). These include two ISVch4 elements of the IS3 family, which have previously been identified as fusion sites in *Vc* strains which fused upon artificial deletion of the DNA adenine methylase Dam [Val et al.] and in a naturally fused *V. cholerae* O1 strain which was isolated from a patient who travelled to Indonesia in 1997 [Yamamoto et al.], highlighting its propensity to serve as fusion site.”

We extracted the following sequences described by Xie et al.¹ to be directly involved in homologous recombination as well as the IS3 elements as amino acid sequences from the published fused chromosome genomes [NCSV1 (NZ_CP010811.1) and NCSV2 (NZ_CP010812.1)].

Strain	Locus	Function
NSCV1	VAA049_1594	HR
NSCV1	VAA049_2432	HR
NSCV1	VAA049_2433	IS3
NSCV2	VAB027_307	HR
NSCV2	VAB027_1228	HR
NSCV2	VAB027_276	IS3
NSCV2	VAB027_1254	IS3

We compared these to HS1 using tblastn and found no matches with a sequence identity >60% or a query coverage >37%, suggesting that these do not encode the same proteins. We added the description above to the Methods section and the following sentence to the manuscript:

“These are not the same recombination sequences and IS3 elements that have previously been described in two naturally fused non-O1 strains [Xie et al.]”

3. Line 142: What is crtS and why is its location important?

Refer to the study by Val et al. (2016) (doi: 10.1126/sciadv.1501914) where the importance of crtS location in the timing of Chr2 replication is shown. When crtS is deleted, chromosome fusion occurs spontaneously as an intermediate before fission (see point 4).

Response:

We added the following sentence including the mentioned reference:

“Previous studies showed that the two *Vc* chromosomes fuse temporarily upon deletion of *crtS* [doi: 10.1126/sciadv.1501914; <https://doi.org/10.1038/s41467-024-55598-9>]”

Figure 2A. The schema of Chr1-Chr2 fusion is rather too simple. Locate at least the origins of replication, the crtS site and some landmark regions (e.g. virulence-associated genomic islands VPI-1, VPI-2, VSP-I and VSP-II).

Response:

We added the loci as suggested and adapted Fig. 2A.

Response:

4. Stability of fused chromosomes:

The authors suggest that fused chromosomes can remain stable for ~12-22 years based on substitution rates. However, rapid fusion/fission events observed within the same household or patient indicate a dynamic process needing further investigation.

Not being an expert in phylogeny, I can't assess the accuracy of the phylogenetic tree or the statistical calculations used to estimate the stability period.

Response:

The ancestral state reconstruction which we used to estimate the stability of chromosome fusions was based on a phylogenetic tree inferred with RaxML (model GTRCAT). While working on these reviews, we realised that this method was not the ideal choice to build the phylogeny of our genomes, as these are very closely related and RaxML assigns a minimal branch length, even for sequences which are the same. We therefore inferred a new

phylogeny using IQtree which allows zero branch length for identical sequences, which we consider to be more reliable (new Figure 2C). Using the same approach of ancestral state reconstruction on the IQtree phylogeny gave us unconvincing results (e.g. assigning an ancestor node as 'fused' with no fused-chromosome descendants), probably due to the small number of variable sites. We therefore removed the ancestral state reconstruction analysis, which we believe contained too much uncertainty and did not add substantially to our conclusions. Instead, we rely on a simpler reading of the phylogeny, which shows at least four independent fusion events on distinct branches (detailed ancestral reconstruction algorithms are not necessary to see this). However, the clades descending from these 5 events are very closely related (often identical genomes within the same patient or household), making it difficult to infer the timing of fission events that occur post-fusion, other than noting the possibility that fusion and fission events occur within a patient (therefore on time scales of a few days). In addition to the updated phylogeny, we have added a new panel 2D which highlights the 5 likely fusion events, and mention these as follows in the Results:

“The phylogeny indicates five potential fusion events, defined as closely related genomes including at least one with a fused chromosome, collected from the same household. Four of these five events are likely independent because they are separated on the phylogeny by other well-supported clades with two chromosomes (Fig 2C; ultrafast bootstrap approximation support >95). Event 1 is a single fusion event in a household contact, with no evidence of onward transmission (Fig 2D, Fig S4). In event 5, 7/10 isolates from the index case had a fused chromosome, but 0/10 isolates from the household contact were fused, which appear to be only distantly related to the index case isolates (12-29 SNVs; Fig S4), again suggesting a single fusion event and no onward transmission. In events 2-4, we observe fused chromosomes amongst the closely related genomes (typically 0-2 SNVs; Fig S4) isolated from both index cases and household contacts, suggesting transmission of the fusion. The clade containing event 2 includes an index case (with 1 fused chromosome out of 9 genomes sequenced) and contact (5/10 fused) from household P, as well as non-fused chromosomes from household O. This observation is consistent with a variety of scenarios, including a single fusion event within household P, followed by transmission of a mixed fused/non-fused population within the household. Alternatively, only non-fused Vc could have been transmitted, followed by independent fusion events in the index and contact within household P. The clades containing events 3 and 4 each involve a single household with one index and two contacts, all of which contained fused chromosomes (Fig 2D). The isolates from these two households are closely related and separated by a low-supported branch of non-fused isolates (ultrafast bootstrap approximation support <25). This is consistent with either independent fusion events in each household, or a single fusion event stable enough to be transmitted between households. Within each household, these clades are also consistent with transmission of a mixed fused/non-fused population, or with 2-3 independent fusion events. A more parsimonious scenario would be a single fusion event in the index case, followed by fused-chromosome transmission to both contacts, and a single fission event in one contact. Although we cannot formally distinguish between these scenarios, it is clear that chromosome fusion occurs repeatedly and is stable enough to be transmitted within, and possibly between households.”

To experimentally measure the stability of chromosome fusion, we passaged two pairs of isolates (one fused and one non-fused isolate from the same patient) for ~200 generations (20 days) in liquid culture and sequenced single colony picks of multiple time-points. We found that all isolates remained stable in their number of chromosomes during this experiment, with no evidence for fusion or fission events under these conditions or time scales (new Fig S5, Table S3).

- The study could clarify whether the fission events resolved from the same HS1 element or whether other mechanisms are involved that could promote genetic exchange between the two chromosomes.

Response:

We investigated whether fission events lead to genetic exchange by comparing the flanking sequences of HS1 in genomes with two assembled chromosomes. If fission events resolve at loci other than HS1, this would lead to sequence exchange at the flanking HS1 (see Figure below).

We extracted the four flanking sequences of HS1 (500bp up and downstream) from all strains which were not fused and carried HS1 on chromosome 1 (n=407) and chromosome 2 (n=349).

All except four genomes had the same four flanking sequences with average nucleotide identities > 99.5%, suggesting no sequence transfer. We examined the four genomes with differing flanking sequences in more detail by comparing them to closely related genomes using blastn via TNA⁴. We found no evidence for DNA transfer upon fusion:

- 103Vc05 (differing flanking sequences 3 and 4): large inversion on chromosome 2 including HS1
- 108Vc03 (differing flanking sequences 3 and 4): Sequence rearrangement just upstream of HS1, confined to chromosome 2 (same as 109Vc05)
- 109Vc05 (differing flanking sequences 3 and 4): Sequence rearrangement just upstream of HS1, confined to chromosome 2 (same as 108Vc03)
- 112Vc05 (differing flanking sequences 2, 3 and 4): Lots of rearrangements on chromosome 1 and 2. The sequenced reads of this isolate assembled in two chromosomes, although we identified reads spanning HS1 and being anchored in both chromosomes (see Fig. S3) and might represent a false negative fusion in the assembly.

In summary, we found no evidence for DNA transfer between the chromosomes upon fission, which indicates that fission occurs at the same site as fusion.

- Chromosome fusions have previously been shown to be transient. In Val et al. (2016), using crtS mutants, chromosome fusions revert to a two-chromosome configuration over 200 generations after acquiring compensatory mutations. This suggests that fusions are generally not stable without specific selective pressures. The author could test the capacity of some of their fused strains (e.g. 111Vc05, 112Vc01...) to revert to two chromosomes under lab conditions. Some of these strains might be blocked in the fused states due to mutations that prevent reversion. Passaging these strains for 200 generations and performing PFGE should suffice to explore this possibility. This experiment would strengthen the assertions made in the manuscript.

Response:

As described above, to measure the stability of chromosome fusion, we passaged two pairs of isolates (one fused and one not fused isolate, each collected from the same patient) for ~200 generations (20 days) and sequenced isolates at day 0, 3, 8, 11, 16 and 20. We observed no evidence for fusion or fission [new Fig S5, Table S3 (see above)]. This suggests that our naturally fused isolates are more stable than the lab-selected fusions from Val et al ⁵.

5. Line 130-135: VSP-I as a potential fusion site:

While VSP-I was identified as a potential homologous sequence for fusion, no evidence was found to support its involvement in the chromosome fusions observed. The manuscript could benefit from discussing why VSP-I might not be used or selected. Would the resulting fusion be non-viable, or viable but not advantageous, and therefore not selected?

Response:

Unfortunately, with the data analysed in this project we cannot know whether a fusion at VSP-I is viable or comes with a fitness defect or not. Not seeing any fusion at VSP-I despite its presence on both chromosomes could suggest that a fusion at VSP-I (a) happens but is not

viable or is selected against; (b) happens but these strains were not sequenced or (c) does not happen or is very unlikely to happen.

We communicate this uncertainty by adding the following:

“Although VSP-I might serve as a potential fusion site, we currently lack evidence for this as none of the fused-chromosome genomes carries more than one VSP-I copy. **Whether fusion at VSP-I takes place and is viable therefore remains unknown.** “

Minor points:

Figure 1C – PFGE : What is the smaller band observed in all samples?

Response:

The band at 1 Mbp is clear for the isolates with non-fused chromosomes (representing the smaller chromosome 2) and fuzzy and weak for the isolates with a fused chromosome. The fuzzy band also occurs in our control *S. pombe*, which does not have a chromosome of that size. We believe the fuzzy band to arise from fragmented DNA (< 1 Mbps) which ran to this position in the pulsed field gel. In PFGE, DNA smaller than a certain size runs at a position near the bottom (potentially the fuzzy 1 Mbps we observe) or off the gel, and DNA bigger than a certain size runs at a position near the top of the gel, or remains in the well; in between there is size-dependent separation (the clear bands arising representing the chromosomes) (personal communication with Prof. David Schwartz, University of Wisconsin, inventor of PFGE⁶). Due to the absence of a 3 Mbp band accompanying the weak 1 Mbp fuzzy band, we do not think the 1 Mbp band arises due to fission of a 4 Mbp chromosome into two chromosomes.

We added the following sentence:

“Some of the fused isolates have a weak, **poorly defined** band at about 1 Mbp in addition to the band at 4 Mbp, but the lack of a band at 3 Mbp in these isolates indicates that fusion did occur. **We hypothesize the weak band to arise from fragmented DNA (< 1 Mbps) which ran to this position in the pulsed field gel.**”

Line 145: In this study, Vc was not “engineered.” The chromosomes fused spontaneously and were selection upon dam depletion.

Response:

Adjusted as follows:

“Finally, chromosome fusion can be **selected experimentally** through depletion of the DNA adenine methylase Dam²”

Reviewer #2 (Remarks to the Author):

Here the authors report evidence of chromosome fusion in *Vibrio cholerae* isolated from clinical cases in Bangladesh. In several of the households studied, chromosome fusion was detected in both cases and contacts, suggesting that *V. cholerae* with a single chromosome can be transmitted. Results were validated using manual assembly and pulsed-field gel electrophoresis to ensure the results were not an artefact. These results are highly novel and

will have significant impact on our understanding of the biology of *Vibrio cholerae*. It is not yet clear what the clinical relevance of such a finding is and this will be an interesting avenue of further research.

Response:

We thank the reviewer for noting the novelty and impact of our work.

Line 53: Is there any data to investigate whether there is an association between fusion and a clinical phenotype, or a participant characteristic such as occupation?

Response:

The isolates with a fused chromosome were collected from ten different participants, six of which experienced Cholera symptoms and four asymptomatic participants. We have added this information to Table S1 and the following to the main text:

“These fused chromosomes were identified in ten different cholera patients from five different households. **These patients include four index cases and six household contacts, two of which experienced cholera symptoms (Table S1).**”

Due to the relatively small number of participants, we refrain from drawing conclusions regarding any association between fusion and clinical outcomes or symptoms.

We further compared two pairs of closely related isolates (collected from the same patient), of which one had a fused chromosome and one two chromosomes. These paired strains were compared in terms of (i) growth, (ii) expression of virulence factors (*ctxA* and *tcpA*) and (iii) their capability to form biofilm formation and found no evidence for an impact of fusion on these phenotypes (new Figure S2).

Line 69: If there is a heterogenous population, some with fusion and some without, why would there not be a band at 3Mbp?

Response:

The isolates subjected to PFGE did not contain a mixed population. The tested isolates represent single colony picks, and sequencing reads for these suggest a homogeneous fusion state. In our dataset we identified one isolate with reads suggesting a heterogeneous fusion state (112Vc05, see Figure S3), which, however, we did not test via PFGE. In future experiments, it would be interesting to subject this isolate to PFGE and passaging experiments to assess stability.

The band at 1 Mbps is clear for the isolates with non-fused chromosomes (representing the smaller chromosome) and fuzzy and weak for the isolates with a fused chromosome. The fuzzy band also occurs in our control *S. pombe*, which does not have a chromosome of that size. We believe the fuzzy band to arise from broken DNA (< 1 Mbps) which ran to this position in the pulsed field gel. In PFGE, DNA that is smaller than a certain size runs at a position near the bottom (potentially the fuzzy 1 Mbps we observe) or off the gel, and DNA bigger than a certain size runs at a position near the top of the gel, or remains in the well; in between there is size-dependent separation (the clear bands arising representing the chromosomes)

(personal communication with Prof. David Schwartz, University of Wisconsin, inventor of PFGE⁶).

We added the following sentence:

“Some of the fused isolates have a weak and smudgy band at 1 Mbp in addition to the band at 4 Mbp, but the lack of a band at 3 Mbp in these isolates indicates that fusion did occur. We hypothesize the smudgy band to arise from fragmented DNA (< 1 Mbps) which ran to this position in the pulsed field gel.”

1: Schwartz and Cantor, *Cell*, 1984, doi: 10.1016/0092-8674(84)90301-5

Line 84/Figure 2A: Would it be possible to indicate on the schematic where HS1 is in relation to the origins of replication? Does the fusion bring VPI-2 closer to an origin of replication, and could this affect its expression?

Response:

We added the ori, crtS and the virulence islands to Figure 2A.

As the fusion site HS1 is part of VPI-2, VPI-2 gets split in the process of fusion which brings the VPI-2 sequences downstream of HS1 closer to ori2 than it is to ori1 in the non-fused state (see Figure 2A). This could affect the expression of genes on VPI-2, depending on activity of ori 1 and 2. Previous studies indicate that the arrangement of loci observed in our fused isolates is associated with both ori sites being active¹. Measuring the activity of ori1 and ori2 as well as the expression of VPI-2 would indeed be interesting future experiments.

We mention our uncertainty on phenotypic effects in the manuscript:

“The clinical or phenotypic consequences of fusion, if any, remain to be explored.”

Line 86/Figure 2B: Adding the corresponding locus tags (e.g. VC1757) for the genes in VPI-2 and chromosome 2 would make it clearer to readers where the homologous sequences are in the genome. What is the genomic context of HS1 in chromosome 2?

Response:

We added the VC locus tags for HS1 to Figure 2B and to the text:

“On chromosome 1, HS1 is located within VPI-2 and **contains** genes **VC1788-VC1803** whereas on chromosome 2 **HS1 was inserted between VCA0451 and VCA0453**. HS1 **encodes** multiple proteins linked to horizontal gene transfer (Fig 2B).”

Line 95/Figure 2C: It would make the figure clearer to have fusion events inferred by ancestral reconstruction shown in Figure 2C, for example by coloring the branches.

Response:

As suggested, we have highlighted the 5 likely fusion events in the phylogeny (updated Fig 2C) and give further detail on these events in the new panel 2D.

The ancestral state reconstruction which we used to estimate the durability of chromosome fusions was based on a phylogenetic tree compiled by RaxML (model GTRCAT). While working on these reviews, we realised that this method was not the ideal choice to build the phylogeny of our genomes, as these are very closely related and RaxML assigns a minimal branch length, even for sequences which are the same. We therefore compiled a new phylogeny using IQtree which indicates zero branch length for the same sequences (new Figure 2C). Using the same approach of ancestral state reconstruction on the IQtree phylogeny gave us unconvincing results (eg. assigning an ancestor node as 'fused' with no fused-chromosome descendants), probably due to the small number of few variable sites. We therefore remove the ancestral state reconstruction analysis.

However, ancestral state reconstruction is not necessary to identify at least four independent fusion events, separated by well-supported non-fused clades on our phylogeny. These events are now indicated in red and number on the updated phylogeny (Fig 2C).

Line 100: By ancestral reconstruction the authors detect 10 fusion and 17 fission events. However as samples are taken from household studies, many pairs of samples will be extremely closely related if not identical. The trees shown in Figure 2 and S3 are both bifurcating, and in Figure 2 branch lengths of length zero are illustrated with a small minimum branch length while S3 is a cladogram. It is therefore not possible to tell from these figures which samples are identical. Could the high number of fusion and fission events in the tree be an artefact of this? Within a household, what is the SNP distance between fused and two-chromosome samples?

Response:

As mentioned above we have inferred a new phylogeny where identical sequences are more sensibly assigned a branch length of zero (Fig 2C). For each household where at least one fusion was observed, we have further determined whole genome SNV differences between genomes, further confirming they are often nearly identical within a household (Fig S4).

We define the following criteria to identify independent fusion events from the phylogeny: Closely related isolates including at least one with a fused chromosome, collected from the same household and interspersed by isolates with two chromosomes. Applying these criteria to the new phylogeny identifies at least four independent fusion events, which we have indicated in the new Figure 2C. In all five households with a fusion observed, there were non-fused genomes with zero SNVs (Fig. S4).

We added the following, detailing the fusion events:

“The phylogeny indicates five potential fusion events, defined as closely related genomes including at least one with a fused chromosome, collected from the same household. Four of these five events are likely independent because they are separated on the phylogeny by other well-supported clades with two chromosomes (Fig 2C; ultrafast bootstrap approximation support >95). Event 1 is a single fusion event in a household contact, with no evidence of onward transmission (Fig 2D, Fig S4). In event 5, 7/10 isolates from the index case had a fused chromosome, but 0/10 isolates from the household contact were fused, which appear to be only distantly related to the index case isolates (12-29 SNVs; Fig S4), again suggesting a single fusion event and no onward transmission. In events 2-4, we observe fused chromosomes amongst the

closely related genomes (typically 0-2 SNVs; Fig S4) isolated from both index cases and household contacts, suggesting transmission of the fusion. The clade containing event 2 includes an index case (with 1 fused chromosome out of 9 genomes sequenced) and contact (5/10 fused) from household P, as well as non-fused chromosomes from household O. This observation is consistent with a variety of scenarios, including a single fusion event within household P, followed by transmission of a mixed fused/non-fused population within the household. Alternatively, only non-fused *Vc* could have been transmitted, followed by independent fusion events in the index and contact within household P. The clades containing events 3 and 4 each involve a single household with one index and two contacts, all of which contained fused chromosomes (Fig 2D). The isolates from these two households are closely related and separated by a low-supported branch of non-fused isolates (ultrafast bootstrap approximation support <25). This is consistent with either independent fusion events in each household, or a single fusion event stable enough to be transmitted between households. Within each household, these clades are also consistent with transmission of a mixed fused/non-fused population, or with 2-3 independent fusion events. A more parsimonious scenario would be a single fusion event in the index case, followed by fused-chromosome transmission to both contacts, and a single fission event in one contact. Although we cannot formally distinguish between these scenarios, it is clear that chromosome fusion occurs repeatedly and is stable enough to be transmitted within, and possibly between households.“

Line 103: What does subsequent mean in this context? Is it the relationship between a node inferred to represent a fission event, and a parent node also inferred to represent a fission event? Or does it refer to the time that samples were collected (e.g. if you were to detect a fission event in a household on day 2, then another on day 9)? Would you not need to calculate the distance between a fusion and subsequent fission event to estimate the stability of the fusion, rather than between two fission events?

Response:

Yes, we previously had calculated the distance between a fusion and subsequent fission events. ‘Subsequent’ in this context refers to a fission event at a node which is a child of a fusion node. We removed the ancestral state reconstruction, and therefore, this sentence no longer appears in the manuscript.

Line 132: Assuming there is no duplication, would I be right in thinking both two-chromosome and fused-chromosome *V. cholerae* contain one copy of HS1 in VPI-2, one copy of HS1 in chromosome 2, and one homologous region in VSP-I? If so, why would the lack of VSP-I copies in the genome preclude it as a recombination site?

Response:

Yes, this statement is correct. The B2 genomes sequenced for this study carry two copies of HS1 (on each chromosome if not fused) and one copy in VSP-I. In our analysis of publicly available genomes, we additionally detect genomes which carry two copies of VSP-I, one on each chromosome. We therefore hypothesize that VSP-I could potentially also serve as fusion sites, but do not detect any fused chromosomes with more than one copy of VSP-I. Why fusions appear to happen at HS1, but not at VSP-I, remains to be clarified.

Table S1: What unit is median read length in? Please add accession numbers.

Response:

The unit is base pairs, which we have now added to the column name. The median read length in our dataset is relatively small (range: 59-6577), resulting from a large number of short reads. However, the N50 of our reads (describing the minimum length of a reads which together represent 50% of data) is relatively high (range: 1,650-19,230), which enables flye to assemble circular chromosomes.

Reviewer #3 (Remarks to the Author):

In this article titled “Prevalent chromosome fusion in *Vibrio cholerae* O1” Cuenod et al report (ad verbatim) “chromosome fusions in clinical *V. cholerae* O1 isolates, including several independent fusion events stable enough to be transmitted between patients within a household. Fusion occurs in a 12 kilobase-pair homologous sequence shared between the two chromosomes, which may lead to reversible chromosomal fusion”. The authors claim, “together, these results show that chromosome fusion via homologous recombination is more prevalent and potentially more stable than previously thought. The clinical or phenotypic consequences of fusion, if any, remain to be explored. This study reveals chromosome fusion in clinical *V. cholerae* O1 strains at an unprecedented scale and highlights the power of long read sequencing to identify structural variation in bacterial genomes.”

While it is true and known that long read sequencing is useful in revealing these types of large structural variations, the other claims of this manuscript are not fully supported. As the authors noted, single chromosome *Vibrio cholerae* (SCV) have been reported in clinical and non-clinical environmental isolates. In fact, similar mechanisms leading to chromosome fusion via homologous recombination have been described in this paper (<https://doi.org/10.1155/2017/8724304>). Did the authors check if they are the same elements or locations involved in those fusion events to rule out that the events described here (12 kb homology in all 58 SCV isolates) are different or different from the Yamamoto et al strain sequence (reference 8)? This leads this reviewer to wonder what is unique in this study.

Response:

We compared the HS1 sequence (described in our manuscript) to the previously published recombination sequence by Xie et al., Val et al. and Yamamoto et al. We found that indeed the IS3 elements on HS1 are the same as described by Val et al. and Yamamoto et al. To clarify this we added the VC locus tags to Fig. 2B and added the following to the main text of this manuscript:

“On chromosome 1, HS1 is located within VPI-2 and **contains** genes **VC1788-VC1803** whereas on chromosome 2 HS1 was inserted between **VCA0451** and **VCA0453**. HS1 **encodes** multiple proteins linked to horizontal gene transfer (Fig 2B). **These include two ISVch4 elements of the IS3 family, which have previously been identified as fusion sites in *Vc* strains which fused upon artificial deletion of the DNA adenine methylase Dam [Val et al.] and in a naturally fused *V. cholerae* O1 strain, isolated in a patient who travelled to Indonesia in 1997 [Yamamoto et al.], highlighting its propensity to serve as fusion site.**“

Although the fusion mechanism described by Xie et al. (we added this reference) also involves an IS3 element, these are not the same sequences as encoded on HS1 described in this manuscript.

To compare the two sequences, we extracted the following annotated genes described by Xie et al. to be directly involved in homologous recombination as well as the IS3 elements as amino acid sequences from the published fused chromosome genomes [NCSV1 (NZ_CP010811.1) and NCSV2 (NZ_CP010812.1)].

Strain	Locus	Function
NCSV1	VAA049_1594	HR
NCSV1	VAA049_2432	HR
NCSV1	VAA049_2433	IS3
NCSV2	VAB027_307	HR
NCSV2	VAB027_1228	HR
NCSV2	VAB027_276	IS3
NCSV2	VAB027_1254	IS3

We compared these to HS1 using tblastn and found no matches with a sequence identity >60% or a query coverage >37%, suggesting that these do not encode the same proteins. We added the description above to the Methods section and the following sentence to the manuscript:

“These are not the same recombination sequences or IS3 elements that have previously been described in two naturally fused non-O1 strains [Xie et al.]”

The authors claim the fusion events are much more prevalent and stable enough to be transmitted within households. This reviewer is not

convinced that these are independent events. How do the authors conclude that these are independent events (all occurring at the same location by the same homology)? Is this conclusion based on phylogenetic diversity of fusion strains? Do they not see the same level of diversity in unfused strains? What is more critical for the prevalence argument is this: 58 out of 467 or 458 (12%) to be precise, may appear to be a high proportion compared to the 2 out of 91 (2.1%) fusion events found in naturally occurring environmental non O1/non O139 strains (<https://doi.org/10.1371/journal.pone.0120311>). In that study, they were shown to be independent events (fusions locations and elements are different). Hence, it is possible that these 58 are probably clonal expansion from single event rather than independent events. This hypothesis (clonal expansion) makes sense given the close spatio-temporal proximity and association of the patients and isolates. Perhaps, sensu lato, I would argue that there may have been ‘5 events’ in 5 households and those clonal isolates got transmitted between patients in that household. On the opposite spectrum, sensu stricto, it might have been a single rare event in the whole site (village) where the samples were collected. Compared to other outbreaks, the SCV was probably prevalent in this outbreak and there was a mix of two chromosome Vibrio (TCV) and SCVs causing the outbreak. Unless the clonality issue is unequivocally resolved, (in TCV vs SCV isolates), the claim that all 58 are independent events is not substantiated.

Response:

Thank you for raising these points requiring clarification. To address the question of how many times fusion happened along the evolution of the strains analysed in this study, we previously performed ancestral state reconstruction [a method to estimate a trait (eg. fusion state) of

hypothetical ancestors in a phylogeny (nodes), by comparing the genomes and traits of the tip of the tree], yielding 10 independent fusion events (not 58).

This ancestral state reconstruction was based on a phylogenetic tree compiled by RaxML (model GTRCAT). While working on these reviews, we realised that this method was not the ideal choice to build the phylogeny of our genomes, as these are very closely related and RaxML assigns a minimal branch length, even for sequences which are the same. We therefore compiled a new phylogeny using IQtree which indicates zero branch length for the same sequences (new Figure 2C). Using the same approach of ancestral state reconstruction on the IQtree phylogeny gave us unconvincing results (eg. assigning an ancestor node as 'fused' with no fused-chromosome descendants), probably due to the small number of variable sites. We therefore removed the ancestral state reconstruction analysis from the manuscript.

We now define the following criteria to identify independent fusion events from the phylogeny: Closely related isolates including at least one with a fused chromosome, collected from the same household and interspersed by isolates with two chromosomes. Applying these to the new phylogeny identifies at least four phylogenetically independent fusion events, which we have indicated in the updated Figure 2C, and detailed in the new panel 2D.

In a scenario of a single fusion event followed by clonal expansion, the fused chromosome strains would all descend from the same branch, which is clearly not the case. The fused chromosome genomes lie on distinct branches of the tree and are interspersed by clades (with high bootstrap approximation support) with two chromosomes, rendering a scenario of a single clonal expansion highly unlikely.

Figure 2 somewhat addressed this question. It is not clear where the single chromosome strains are in this tree? Do they cluster or dispersed throughout the tree? The heat map indicated most SCV isolates were from 2 households (green and purple on top).

Response:

The heatmap is aligned to the tree tippoints; it is therefore apparent that the fused chromosome strains are clustered on different branches of the tree. To make the visualization more clear, we have indicated and numbered in red the 5 clades containing fusions, and highlighted the branches where fused-chromosomes are found.

It would really help if a simple table capturing all the isolate metadata, as shown in the below example with these headers, is presented.

Household # (H) Patient # (P) Type [(index I/Sec (S)] Total sequenced Fused Bipartite Colony # H/P/T

The last 2 columns info for all 58 SCV colonies. This table would inform the reader on whether there is bias and clonal expansion or not.

Response:

We have added the columns 'Individual' (participant identifier), 'Household' (household identifier), 'Type participant' (index case or household contact) and 'Symp' (whether the participant experienced symptoms or not) to Table S1, adding the suggested information.

The second major criticism is this: What is not clear is that how the fusions are stably maintained without 'fission'. That is, how are they locked in the SCV configuration? I would expect an equilibrium of fused and separated chromosomes in a population of bacteria in vivo in patients as well as in vitro (in cultures grown for gDNA extraction) unless the SCV is recombination deficient after fusion. Did the authors check if the bacteria have intact *recA*, for example, or if the bacteria are recombination proficient by other phenotypes; for example, UV resistance or spontaneous mutation frequencies resulting from homologous recombination? Both are easy phenotypes to check. The authors may argue that these are not focus of the manuscript. But these are critical questions that need to be addressed to make this a unique study with significant new findings.

Response:

Many thanks for raising this important point. We agree that an equilibrium of fused and non-fused chromosomes is a probable scenario, whose dynamics are not yet well understood. We make the distinction between 'stability' (a relative term, implying stability on some time scale) and 'locked' configurations in which fusion is permanently stable. We make no claims about permanent (locked) fusion, only about relative stability over certain time scales (in our case, over 200 generations of a passage experiment or a few days for patient-to-patient transmission within a household).

To test the stability of chromosome fusion under laboratory conditions, we passaged two pairs of isolates (each pair consisting of one fused and one non-fused isolate), each from the same patient for 200 generations and sequenced single colony picks at several time points. We found:

- The chromosome state (fused or bipartite) to be stable during this experiment on the level of assemblies (Table S3) and sequence reads (Figure S5)
- As suggested by the reviewer, we asked whether spontaneous mutation frequencies were different in fused-chromosome isolates, as might be expected in *RecA*-deficient mutants. After 200 generations we found no apparent differences in mutation frequencies between the fused and the non-fused chromosomes, suggesting no difference in recombination proficiency upon fusion (Figure S6).

In terms of patient-to-patient transmission, we observed 3 households (numbered 2-4 in the phylogeny, Fig 2C) in which fusions are common in 2 or 3 different patients from the same household. This implies that fused chromosomes are likely stable enough to transmit between patients (likely on time scales of a few days). The alternative is that the lineage infecting this household is prone to repeated fusion and un-fusion events. In the updated text, we mention these alternative scenarios but deem it likely that there is at least some transmission of fused chromosomes occurring, possibly in a mixed population with both fused/non-fused chromosomes.

To examine whether the fused strains are recombination deficient, we performed the following bioinformatic analysis:

- We investigated all *mut* and *rec* genes and found no SNV occurring exclusively or being enriched in fused-chromosome strains.
- Broadening our search for fusion specific SNVs (which could help explain the stability of the fused chromosome), we ran a simple genome wide association study (pyseer).

We found no protein-coding sequence variation significantly associated with chromosome fusion.

In summary, this data suggests that chromosome fusion is stable under certain laboratory conditions, without evident recombination deficiency. Exploring the mechanisms of chromosome fusion stability in future experiments will be very interesting, as there likely exist some conditions in which fusion is unstable. Future research is further needed to explore whether chromosome fusion/fission is in balance within patients, or if not, what stabilizes the respective fusion states.

To clarify, we added the following to the main text:

“To further quantify the stability of chromosome fusion, we conducted a passaging experiment and found the fusion state to be stable over 200 generations under laboratory conditions (Fig. S5, Table S3). This apparent stability could be explained if fused isolates are recombination-deficient and therefore unable to reverse the fusion event. However, this does not appear to be the case: genes involved in recombination (all *rec* and *mut* genes) did not contain any fusion-specific mutations. To search for other mutations that could potentially stabilize chromosomal fusion, we used a genome-wide association study (Methods), but did not find any protein-coding variants significantly associated with fusion state. Nor did we identify any significant differences in the number of SNVs accumulated during the passage experiment between fused and unfused isolates (p -value >0.05 , Mann-Whitney U), which would have been expected had they differed in their recombination or DNA repair proficiency (Fig. S6). In conclusion, we find no evidence that fused isolates are recombination deficient. Yet, we detected closely-related fused and non-fused isolates collected from the same household and patient, suggesting fusion/fission events can occur within patients, likely on time scales of a few days. How chromosome fusion is stabilized under laboratory conditions and whether fusion/fission events are more dynamic within patients remains to be examined.”

There are additional minor comments throughout the manuscript that might improve/clarify certain part of the manuscript. Please see the edited version.

Please find point by point answers below.

Line 31: SCVs have been reported in Vc o1 as well as non O1 non O139 isolates. What is unique in this report?

Response:

Whereas previous studies report isolated events of chromosome fusion (Val et al., Xie et al.), which in one case was unstable under laboratory conditions (Val et al.), we observe multiple independent events of chromosome fusion in circulating clinical strains which is stable for ~200 generations under laboratory conditions. Our report is therefore unique in identifying multiple independent fusion events, with quantifiable stability.

Line 32: How do you know they are independent and non clonal expansion? Do these represent different fusion locations?

Response:

As described in detail above, the fused-chromosome genomes appear on different branches in the phylogenetic tree, suggesting at least four different fusion events. These were sampled from different households, further corroborating the hypothesis of independent fusion events. The independent events are interspersed by other well-supported non-fused branches in the phylogeny, supporting that the fusions involve at least four independent events.

Line 35: How does it compare to earlier reports in terms of location of fusions?

Response:

We compared HS1 to the fusion sites previously described by Val et al., Xie et al. and Yamamoto et al. and added this to the manuscript (see response to comment above).

Line 35: 'Reversible chromosome fusion (abstract)': Do you see reversion?

Response:

We hypothesize the observed fusion to be reversible, as there are closely related fused and non-fused genomes in our dataset, suggesting a dynamic fusion state. As we do not directly observe fission, we adjusted the sentence as follows:

"Fusions occur in a 12 kilobase-pair homologous sequence shared between the two chromosomes and are stable for 200 generations under laboratory conditions"

Line 38: '7PET': IS this a commonly used abbreviation?

Response:

Yes, it is becoming a standard abbreviation. See for example: doi:[10.1038/s41467-020-19185-y](https://doi.org/10.1038/s41467-020-19185-y), <https://doi.org/10.1038/s41564-023-01472-1>, DOI: [10.1126/science.aad590](https://doi.org/10.1126/science.aad590)

Line 56: How do the 58 genomes parse out among the 10 patients? Trying to understand if they are clonal or not?

Response:

As described above, the 58 fused genomes are clearly not derived from a single clonal expansion. Rather, they fall into independent events. This is now clearly indicated on the phylogeny (Fig 2C) in red numbers. We have added the columns 'Individual' (participant identifier), 'Household' (household identifier), 'Type participant' (index case or household contact) and 'Symp' (whether the participant experienced symptoms or not) to Table S1, adding the suggested information.

Line 69 'fused isolates':

- Which ones? there is no discrete band.
- COuld this 1Mb be indication of reversal? there is not enough large chromosome to see ?

Response:

We hypothesize the fuzzy band to arise from fragmented DNA (< 1 Mbps), rather than a heterogeneous population, because we do not observe any evidence of a heterogeneous state in the raw reads of the isolates tested by PFGE (see Fig. S2 (new Figure S3)). In PFGE, DNA that is smaller than a certain size runs at a position near the bottom (potentially the fuzzy 1 Mbps we observe) or off the gel, and DNA bigger than a certain size runs at a position near the top of the gel, or remains in the well; in between there is size-dependent separation (the clear bands arising representing the chromosomes) (personal communication with Prof. David Schwartz, University of Wisconsin, inventor of PFGE⁶).

We added the following sentence:

“Some of the fused isolates have a weak **and smudgy** band at 1 Mbp in addition to the band at 4 Mbp, but the lack of a band at 3 Mbp in these isolates indicates that fusion did occur. **We hypothesize the smudgy band to arise from fragmented DNA (< 1 Mbps) which ran to this position in the pulsed field gel.**”

Line 75: Give a simple flow diagram of the 10 patients and how many colonies sequences per patient and how many were fused and bipartite. Of the 58 how many come from each patient and whether index or secondary case

Response:

We added the required information to Table S1 (see above).

Line 85: One would expect to see recombinant products as well unless the fused configuration is locked somehow by genetic mutations? for example in RecA. Did you look at if they are recombination proficient?

Response:

We did not find any evidence for these strains to be recombination deficient (see response above for more detail).

Line 91: dynamics of fusion or dynamics of transmission? Make it clear.

Response:

As we removed the ancestral state reconstruction from our manuscript, we edited the sentence which now reads as follows:

~~“To investigate the dynamics of chromosome fusion in our samples, we examined the phylogenetic distribution of fused and non-fused states and reconstructed the likely ancestral states along the phylogeny.”~~

Line 127: Cite nonO1-non_O139 examples

Response:

We added the following:

“These results **and previous studies** [6-8] suggest that fusion, while rare, can occur in different *Vibrio* species and *Vc* sublineages.”

Line 146: awkward not our genomes- genomes described here:

We edited as suggested.

Line 151-152: Both these statements are not corroborated: 1) How do you discount clonality? Have you looked at relatedness among the 58 isolates?

Please see above for details on our phylogenetic analysis.

2) How do you know the stability? have you looked at long term growth to see fission events. To measure the stability of chromosome fusion, we passaged two pairs of isolates (one fused and one not-fused isolate), each from the same patient, for 200 generations (20 days) and sequenced a single colony pick for each strain at day 0, 3, 8, 11, 16 and 20. We observed no evidence for fusion or fission (new Fig S4, Table S3).

Bibliography:

1. Xie, G. *et al.* Exception to the Rule: Genomic Characterization of Naturally Occurring Unusual Strains with a Single Chromosome. *Int J Genomics* **2017**, 8724304 (2017).
2. Val, M.-E. *et al.* Fuse or die: how to survive the loss of Dam in *Vibrio cholerae*. *Mol Microbiol* **91**, 665–678 (2014).
3. Yamamoto, S. *et al.* Single Circular Chromosome Identified from the Genome Sequence of the *Vibrio cholerae* O1 bv. El Tor Ogawa Strain V060002. *Genome Announc* **6**, (2018).
4. TNA documentation — TNA 0.0.1 documentation. <https://tna.readthedocs.io/en/v0.2.0/>.
5. Val, M.-E. *et al.* A checkpoint control orchestrates the replication of the two chromosomes of *Vibrio cholerae*. *Sci Adv* **2**, e1501914 (2016).
6. Schwartz, D. C. & Cantor, C. R. Separation of yeast chromosome-sized DNAs by pulsed field gradient gel electrophoresis. *Cell* **37**, 67–75 (1984).